# Ranking Time Series using a Time Warping Ideal Point Model

Lucas Zoroddu [1]   Pierre Humbert [2]   Laurent Oudre [1]

## Abstract

Expert-annotated time series datasets often suffer from low agreement, especially in medical applications where decisions rely on subjective criteria and inconsistent thresholds. Such variability degrades annotation quality and thus limits the reliability of supervised classification models. To address this, we propose to rely on a pairwise comparison-based approach, which provides a more robust alternative to individual annotation, since relative judgments are typically easier and yield higher consistency. The problem is thus transformed into a ranking problem and we introduce an ideal point model adapted to time series data using elastic similarity measures such as Dynamic Time Warping (DTW) and Time Warp Edit Distance (TWED). We prove Lipschitz continuity of these distances and demonstrate several convergence guarantees for this model. To facilitate gradient-based optimization, we also introduce a differentiable version of the TWED. Finally, we show through multiple experiments that our approach produces accurate and robust rankings under noisy annotation conditions.

## 1. Introduction

Time series data arise in numerous scientific and industrial domains, from physiological monitoring to finance and sensor networks. Many real-world applications lead to supervised classification tasks, such as detecting abnormal events in physiological or mechanical signals, or distinguishing between different types of motion. These tasks typically rely on binary annotations provided by experts. However, in several domains, especially in medicine, expert annotations suffer from substantial variability due to subjective interpretation and the difficulty of defining clear decision thresholds. This can result in low inter- and intra-annotator agreement, making it difficult to construct reliable annotated datasets and thus to train reliable supervised models. Expert annotations often rely on subjective internal criteria, not just on explicit decision thresholds. Moreover, as noted in (Sylolypavan et al., 2023), common consensus strategies such as majority voting may still produce suboptimal models, and there may not exist a "super expert" whose labels can serve as a reliable gold standard.

A natural way to address these limitations is to adopt alternative annotation strategies. Pairwise comparisons offer an appealing alternative: instead of assigning absolute labels, annotators only need to decide which of two samples is more abnormal, severe, or relevant. Such relative judgments are often easier and more consistent, leading to higher-quality supervision (Saaty, 2008; Kalpathy-Cramer et al., 2016; Fürnkranz & Hüllermeier, 2010). In medical imaging, for instance, pairwise and sorting-based strategies have been shown to significantly improve inter-expert agreement (Jang et al., 2022; Kakiashvili et al., 2012).

Pairwise annotations naturally shift the learning problem from classification to ranking. Various statistical models have been proposed to infer global rankings from partial comparisons of several items. Among them, *Ideal Point Models* (IPMs) have been widely used in social sciences and machine learning (Coombs, 1950; Massimino & Davenport, 2021; Jamieson & Nowak, 2011). In IPMs, items and annotators are embedded in a latent space, and preferences depend on distances to an ideal point. While existing work has focused on learning these latent ideals in $\mathbb{R}^d$ (Canal et al., 2022), our goal is to adapt IPMs to rank time series. This task is difficult because time series can vary in length and sampling frequency, exhibit distortions or time lags, and be affected by noise or partial misalignment. As a result, vector-based feature integrations often fail to capture their temporal structure, motivating the use of methods that operate directly on time series.

To handle such challenges, distance measures like Dynamic Time Warping (DTW) (Sakoe & Chiba, 1978), Edit Distance on Real Sequences (EDR) (Chen et al., 2005), or the Time Warp Edit Distance (TWED) have been de-

[1]Université Paris Saclay, Université Paris Cité, ENS Paris Saclay, CNRS, SSA, INSERM, Centre Borelli, F-91190, Gif-sur-Yvette, France [2]Université Paris-Saclay, CNRS, Univ Evry, Laboratoire de Mathématiques et Modélisation d'Evry, 91037, Evry-Courcouronnes, France. Correspondence to: Lucas Zoroddu <lucas.zoroddu@ens-paris-saclay.fr>.

*Proceedings of the 43rd International Conference on Machine Learning*, Seoul, South Korea. PMLR 306, 2026. Copyright 2026 by the author(s).

veloped. TWED (Marteau, 2009) in particular is a true metric, combining temporal elasticity and edit operations into a coherent geometric structure. This paper bridges these two paradigms by defining an Ideal Point Model for Time Series Ranking, where the latent geometry is induced by the TWED distance or the DTW between time series. This yields a ranking model that respects temporal structure while retaining interpretability. In particular, by modeling the ideal as an explicit time series rather than as a point in a latent vector space, the annotator's decision process becomes directly observable. This representation makes it possible to analyze which temporal patterns, alignments, or distortions drive the ranking, thereby providing a more transparent and interpretable explanation of the annotator's preferences.

**Contributions:** The main contributions of this paper are:

- Theoretical advances on time-elastic distances, including new properties of DTW and TWED (Lipschitz continuity) and the introduction of a differentiable variant of the TWED measure.

- The introduction of the IPM for time series, together with theoretical guarantees and convergence results.

- Extensive experiments demonstrating the effectiveness of the proposed model.

## 2. Ranking from Pairwise Comparisons

In this section, we introduce the standard models for ranking from pairwise comparisons and detail the Ideal Point Model (IPM), which is central to this article.

### 2.1. Standard Models

The problem of ranking from pairwise comparisons is generally modeled as follows: let us consider $n$ items with an unknown preference matrix $M$, where $M_{ij}$ is the probability that item $i$ is preferred to item $j$, with $M_{ij} + M_{ji} = 1$. Then, (noisy) pairwise comparisons $(i, j, y)$ are observed, where $i$ and $j$ are items and $y \in \{-1, 1\}$ the preference label ($-1$ if $i$ is preferred to $j$). The objective is to retrieve the preference matrix $M$ from these observations.

**Parametric models.** Parametric models are an important family of ranking methods. They are based on the assumption of the existence of latent scores $\{w_i\}_{i=1}^n$ associated to each item, and a strictly increasing link $F$ such that

$$M_{ij} = F(w_i - w_j) .$$

Hence, ordering the $w_i$'s is equivalent to ordering the items. For example, the so called Bradley-Terry-Luce (BTL) model (Bradley & Terry, 1952) assumes that the

probability that item $i$ is preferred to item $j$ is given by:

$$M_{ij} = \frac{1}{1 + \exp(-(w_i - w_j))} .$$

Then, the $w$ are estimated using a maximum likelihood approach; since no closed form exists, gradient-based or Majorize-Minimization algorithms are used (Hunter, 2004). Other well-known models based on this idea include Thurstone (Thurstone, 2017) and Mallows (Mallows, 1957) for example.

**Non-parametric estimators.** In a nonparametric framework, low rank based estimators have been proposed for recovering the matrix $M$ and the associated ranking. Spectral and low-rank methods consider that the observations give a noisy version of $M$ and perform matrix denoising via singular value thresholding (see e.g. (Cai et al., 2008)). Another class of methods works under the strong stochastic transitivity assumption (Fishburn, 1973) and assumes that $M$ belongs to the class of antisymmetric matrices consistent with an unknown permutation. Its estimation can then be done via a least-squares projection problem (Shah et al., 2016). Finally, a simpler but effective alternative consists of counting-based estimators which compute the number of times item $i$ is preferred to the other items and rank all the items according to a particular score (Emerson, 2013; Wauthier et al., 2013).

### 2.2. The Ideal Point Model

While the previous methods leverage only pairwise annotation, some methods in the literature assume that the items are embedded in a metric space ($\mathbb{R}^p$ for instance) and use this embedding as additional information to retrieve the ranking. One can cite RankSVM (Joachims, 2002) or RankNet (Burges et al., 2005). For the latter, it is a variant of the BTL model replacing the latent scores $w_i$ by $f(x_i)$ where $x_i$ is the item embedding and $f$ is a parametric function to optimize. On the other hand, the Ideal Point Model (IPM) (Coombs, 1950) assumes that there is an ideal point and that the ranking is done using the distances to this ideal. Formally, given a distance $d$ on $\mathbb{R}^p$ and an ideal point $u^* \in \mathbb{R}^p$, for $x, x' \in \mathbb{R}^p$, $x$ is preferred to $x'$ if and only if $d(x, u^*) < d(x', u^*)$. A full ranking can thus be retrieved by sorting the items by their distance to the ideal point $u^*$.

Solving this problem and finding $u^*$ from pairwise comparisons highly depends on the chosen distance $d$ and was well studied in the literature for the euclidean distance (Coombs, 1950; Massimino & Davenport, 2021; Jamieson & Nowak, 2011). In this work, we propose to leverage distance metrics adapted to time series, enabling a robust ranking while accounting for time warping deformations.

## 3. Distances between Time Series

To design a relevant ideal point model for time series, we must go beyond generic time series similarity measures. Our objective is to adapt the Ideal Point Model so that the ideal itself is a time series. This requires similarity measures that operate directly on the waveform of the series, rather than on extracted features or aggregated representations. The standard $L^q$ metrics could be used but they are not convenient for comparing time series since they are not robust to time warping deformations. On the contrary, elastic distances can be robust to this type of deformations (see e.g. (Holder et al., 2023)).

In this paper, we focus on two elastic similarity measures for time series: Dynamic Time Warping (DTW) (Sakoe & Chiba, 1978) and Time Warp Edit Distance (TWED) (Marteau, 2009). DTW is the most widely used elastic measure for time series comparison and has long been considered the state of the art for 1-nearest-neighbor classification (Bagnall et al., 2017). Its practical effectiveness stems from its ability to perform nonlinear temporal alignment between sequences, although it does not satisfy the properties of a metric. TWED can be seen as a metric variant of elastic time series distances. By combining temporal warping with edit operations and introducing a stiffness parameter, TWED satisfies the triangle inequality and thus is a proper metric. Consequently, we consider both DTW, for its empirical effectiveness and widespread adoption, and TWED, for its metric structure and theoretical advantages.

### 3.1. DTW Distance

The DTW (Sakoe & Chiba, 1978) computes the best possible alignment between two time series of respective length $l$ and $m$ by computing first the $l \times m$ pairwise distance matrix between these points and then solving a dynamic program (DP).

**Definition 3.1.** Let $X = (x_1, \ldots, x_l)$ and $X' = (x'_1, \ldots, x'_m)$ be two time series, and let $\delta(x_i, x'_j)$ denote the local dissimilarity (or cost) between points $x_i$ and $x'_j$. The DTW cost between $X$ and $X'$ is recursively given by:

$$D_{i,j} = \begin{cases} \delta(x_1, x'_1), & \text{if } i = j = 1 \\ \delta(x_i, x'_j) + \min\{D_{i-1,j}, \\ D_{i,j-1}, D_{i-1,j-1}\} & \text{otherwise ,} \end{cases}$$

for $1 \le i \le l$, $1 \le j \le m$, with the convention $D_{0,j} = D_{i,0} = +\infty$ for boundary initialization.

The DTW distance between $X$ and $X'$ is then given by $\mathrm{DTW}(X, X') = D_{l,m}$.

*Remark* 3.2. The recursive formula is convenient for implementation, but the DTW can also be written in terms of alignments, which can be more convenient (see Appendix

A.1).

In the following, for a time series $X \in \mathbb{R}^{p \times T}$, we denote by $\|.\|_q$ the $L^q$ norm of $X$ considered as a vector, i.e:

$$\|X\|_q = \left( \sum_{i=1}^{T} \sum_{j=1}^{p} |X_{ij}|^q \right)^{1/q} .$$

We now provide a new result on DTW that will be useful later on.

**Proposition 3.3.** *Assuming that* $\delta(x_i, x'_j) = \|x_i - x'_j\|_q$, *for* $X, X', X'' \in \mathbb{R}^{p \times T}$, *the DTW distance satisfies:*

$$|DTW(X, X') - DTW(X, X'')|$$
$$\le \sqrt{T(T-1)} \, p^{max(0, \frac{1}{q} - \frac{1}{2})} \|X' - X''\|_2,$$

*i.e DTW is* $\sqrt{T(T-1)} \, p^{max(0, \frac{1}{q} - \frac{1}{2})}$ *Lipschitz wrt the* $L^2$ *norm. This inequality cannot be improved.*

*Proof.* The proof is given in Appendix A.1. □

### 3.2. TWED Distance

The TWED, introduced in (Marteau, 2009), is a measure of dissimilarity between two time series that combines temporal warping with edit operations.

**Definition 3.4** (TWED[1]). Let two time series be defined as $X = (x_1, x_2, \ldots, x_l)$ and $X' = (x'_1, x'_2, \ldots, x'_m)$ where $x_i, x'_j \in \mathbb{R}^p$ are multivariate samples.

The TWED distance $\delta_{\lambda, \nu}(X, X')$ between $X$ and $X'$ is recursively defined by:

$$D_{i,j} = \min \begin{cases} D_{i-1,j-1} + \delta(x_i, x'_j) + \delta(x_{i-1}, x'_{j-1}) \\ \quad + 2\nu(|i - j|) \\ D_{i-1,j} + \delta(x_i, x_{i-1}) + \lambda + \nu \\ D_{i,j-1} + \delta(x'_j, x'_{j-1}) + \lambda + \nu , \end{cases}$$

with boundary conditions: $D_{0,0} = 0$, $D_{i,0} = \sum_{k=1}^{i} \left( \delta(x_k, x_{k-1}) + \lambda \right)$, $D_{0,j} = \sum_{k=1}^{j} \left( \delta(x'_k, x'_{k-1}) + \lambda \right)$, $\delta(x_0, x'_0) = 0$, where $\delta(x, x')$ is a local distance function (typically the (squared) Euclidean distance), $\nu \ge 0$ is a stiffness parameter controlling sensitivity to time shifts and $\lambda > 0$ is a penalty term for deletion (edit operation). Finally, we define $\delta_{\lambda, \nu}(X, X') = D_{l,m}$.

**Proposition 3.5.** *Assuming that* $\delta(x_i, x'_j) = \|x_i - x'_j\|_q$, *for* $X, X', X'' \in \mathbb{R}^{p \times T}$ *the TWED distance satisfies:*

$$|\delta_{\lambda, \nu}(X, X') - \delta_{\lambda, \nu}(X, X'')|$$
$$\le \sqrt{4T - 3} \, p^{\max(0, \frac{1}{q} - \frac{1}{2})} \|X' - X''\|_2 ,$$

---

[1]Here, the definition is given for constant timestamps. See Marteau (2009) for a more general definition.

*i.e,* $\delta_{\lambda,\nu}$ *is* $\sqrt{4T-3}$ $p^{\max(0,\frac{1}{q}-\frac{1}{2})}$ *Lipschitz wrt the* $L_2$ *norm. This inequality cannot be improved.*

*Proof.* The proof is given in Appendix A.2. □

## 4. Mathematical Formulation

For simplicity, in the following section we only consider univariate time series. The extension of our theoretical results to the multivariate case is postponed to Appendix B.2.

### 4.1. Ideal Point Model

Let us consider a similarity measure $d$ on $\mathbb{R}^T$ and $u^* \in E \subseteq \mathbb{R}^T$, called the ideal time series. For $X, X'$ two time series in $E$, the IPM says that $X$ is preferred to $X'$ if and only if

$$d(X, u^*) < d(X', u^*) . \tag{1}$$

Suppose that we observe a dataset of $n$ i.i.d. samples

$$S = \big((X_1, X_1', y_1), \ldots, (X_n, X_n', y_n)\big),$$

where each sample consists of a pair of time series $X$ and $X'$, along with a label $y \in \{-1, 1\}$ indicating which series is preferred. The goal is thus to recover the unknown ideal point $u^*$ from these observations. To this end, one possibility is to minimize the true risk associated with a candidate time series $u \in \mathbb{R}^T$:

$$R(u) := \mathbb{E}\big[\ell\big(y\big(d(X, u) - d(X', u)\big)\big)\big] , \tag{2}$$

where $\ell : \mathbb{R} \to \mathbb{R}_+$ is a loss function (e.g. the hinge loss). We will denote by $\tilde{u}$ one of the minima of $R(\cdot)$ on $E$ (assuming it exists). Note that $\tilde{u}$ is not necessarily equal to $u^*$ because the labels are not necessarily faithful to the model (e.g. if the labels come from an annotator). This important question will be addressed in Section 4.3.

In practice, the true risk being unknown, we minimize its empirical version:

$$\hat{R}_n(u) := \frac{1}{n} \sum_{i=1}^n \ell\big(y_i\big(d(X_i, u) - d(X_i', u)\big)\big) . \tag{3}$$

The estimated ideal point is then given by (under the existence assumption):

$$\hat{u}_n := \arg\min_{u \in E} \hat{R}_n(u) . \tag{4}$$

**Proposition 4.1.** *If $d$ verifies* $\big(d(x,y) = 0 \implies x = y\big)$, *the ideal time series $u^*$ is unique.*

*Proof.* Let $u_1^* \neq u_2^*$ in $\mathbb{R}^T$ be two ideal points resulting in the same model. $0 = d(u_1^*, u_1^*) < d(u_1^*, u_2^*)$ so

$u_1^*$ is preferred to $u_2^*$. However we also have that $0 = d(u_2^*, u_2^*) < d(u_1^*, u_2^*)$ which is absurd. Thus the ideal point is unique. □

*Remark* 4.2. Note that the TWED distance verifies the assumption of Proposition 4.1, implying a well define ideal point model with a unique $u^*$. However this assumption is not verified by the DTW measure. Indeed, if $u^*$ is an ideal point for the DTW model, then any time warped version of $u^*$ (i.e. any $v \in \mathbb{R}^T$ verifying $DTW(u^*, v) = 0$) is also an ideal point.

### 4.2. Risk Generalization Bounds

**Assumption 4.3.** We assume that $d$ is a similarity measure on $\mathbb{R}^T$ such that for all $u \in \mathbb{R}^T$, $x \mapsto d(u, x)$ is $L_d$-Lipschitz wrt to the $L^2$ norm and the loss function $\ell$ is assumed to be $L$-Lipschitz. Moreover, we assume that $E$ is a compact set and we define $\gamma := \sup_{X, X' \in E} ||X - X'||_2 < \infty$.

**Theorem 4.4.** *Let $S = ((X_1, X_1', y_1), \ldots, (X_n, X_n', y_n))$ a dataset of tuples with time series $X_i$, $X_i'$ in $E$, and labels $y_i \in \{-1, 1\}$. Let $\tilde{u}$ and $\hat{u}_n$ be minimizers of $R$ and $\hat{R}_n$ on $E$, respectively. Under Assumption 4.3, we have that with probability at least $1 - 2\delta$,*

$$R(\hat{u}_n) - R(\tilde{u})$$
$$\leq (l_0 + 3L_dL\gamma)\sqrt{\frac{\log(1/\delta)}{2n}} + \tilde{C}L_dL\gamma\sqrt{\frac{T}{n}} ,$$

*where $\tilde{C}$ is an explicit constant that does not depend on any of the problem parameters and $l_0 = \ell(0)$.*

**Corollary 4.5.** *Let $d$ be the DTW with $\delta(x_i, x_j') = |x_i - x_j'|$. Under the same assumptions as Theorem 4.4, we have that with probability at least $1 - 2\delta$,*

$$R(\hat{u}_n) - R(\tilde{u}) \leq (l_0 + 3\sqrt{T(T-1)}L\gamma)\sqrt{\frac{\log(1/\delta)}{2n}}$$
$$+ \tilde{C}T\sqrt{T-1}L\gamma\sqrt{\frac{1}{n}} .$$

**Corollary 4.6.** *Let $d$ be the TWED with $\delta(x_i, x_j') = |x_i - x_j'|$. Under the same assumptions as Theorem 4.4, we have that with probability at least $1 - 2\delta$,*

$$R(\hat{u}_n) - R(\tilde{u}) \leq (l_0 + 3\sqrt{4T-3}L\gamma)\sqrt{\frac{\log(1/\delta)}{2n}}$$
$$+ 2\tilde{C}TL\gamma\sqrt{\frac{1}{n}} .$$

*Proof.* The proofs are given in Appendix B.1. □

*Remark* 4.7. Note that in the previous bounds, the dimension of the problem $T$ is also hidden in the constant $\gamma$. Indeed, if the time series take values in $E = [-M, M]^T$, then

$\gamma = 2M\sqrt{T}$. Thus, for the DTW, the convergence rate is in $O\left(\frac{T^2}{\sqrt{n}}\right)$, and it is $O\left(T\sqrt{\frac{T}{n}}\right)$ for TWED.

### 4.3. Recovery Guarantees

According to the previous theorem, when the number of samples grows, the risk of the estimated ideal point converges towards the risk of the minimizer $\tilde{u}$ of $R$ (the ideal risk). It remains the important question of the recovery of $u^*$, i.e, does $u^* = \tilde{u}$ and does $\hat{u}_n$ converge towards $u^*$ ? In this section, we study the conditions on the distribution of $(X, X', y)$ to have $u^*$ being the unique minimizer of $R$ and we prove the almost sure convergence of $\hat{u}_n$ towards $u^*$.

We adopt a noise model similar to the one proposed in (Canal et al., 2022).

**Definition 4.8** (Noise model). Let $f : \mathbb{R} \to [0,1]$ be a strictly monotonically increasing link function satisfying $f(t) = 1 - f(-t)$, for example the logistic link $f(t) = (1 + e^{-t})^{-1}$. For a pair of random time series $(X, X')$, define

$$\Delta_{X,X'}(u) := d(X, u) - d(X', u)$$

where $d$ is a distance and

$$P\big(y = -1 \mid X, X'\big) = f\big(-\Delta_{X,X'}(u^*)\big).$$

This formulation captures the intuition that the closer the two time series, the higher the uncertainty. The negative log-likelihood of this distribution is given by:

$$\ell_f(y, X, X'; u) := -\log\big(f(y\,\Delta_{X,X'}(u))\big).$$

**Assumption 4.9.** For all $u \neq v \in \mathbb{R}^T$, the measure of the pairs distinguishing $u$ and $v$ is positive, i.e.:

$$\mathbb{P}\big(\{(X, X') : \Delta_{X,X'}(u) \neq \Delta_{X,X'}(v)\}\big) > 0.$$

We make two remarks here: first, note that if $d$ is not a metric (more specifically if $d$ does not verify $d(X, X') = 0 \implies X = X'$), there is no distribution that satisfies this assumption. Second, if the space is a $L^2$ ball in $\mathbb{R}^T$ (i.e the samples and the ideal are in this ball), any distribution with a positive density function on this ball verifies this assumption.

**Lemma 4.10.** *Assuming data follow the noise model and defining the risk using the loss $\ell_f$ (Definition 4.8). Under assumption 4.9: (i) $u^*$ is a minimizer of $R$ and (ii) this minimizer is unique. Consequently, $u^* = \tilde{u} := \arg\min_u R(u)$.*

**Theorem 4.11.** *Under the same assumptions as Theorem 4.4 and Lemma 4.10, we have $\hat{u}_n \xrightarrow{a.s} u^*$.*

*Proof.* Proofs of these results are given in Appendix C. $\square$

### 4.4. Low Dimension Modeling

The two studied similarity measures (TWED and DTW) can compare two time series with different lengths. The low dimension model is then defined as the previous one but taking $u^*$ in $\mathbb{R}^r$ with $r < T$.

In this section we analyze the generalization of the previous theorems to this low dimension model.

The intuition behind this low dimension model is that for some applications, the ideal signal could be compressed and only a few timestamps could be necessary to summarize it. This part is very related to the problem of signal compression and downsampling (cf Theorem 4 in (Marteau, 2009)). While considering $u^*$ in $\mathbb{R}^r$ with $r \ll T$ could lead to an easier training, it also leads to better Lipschitz constants, thus better generalization bounds. Indeed, in the univariate case, we have that:

$$\textbf{DTW}: \qquad R(\hat{u}_n) - R(\tilde{u}) = O\left(\frac{Tr}{\sqrt{n}}\right)$$
$$\textbf{TWED}: \qquad R(\hat{u}_n) - R(\tilde{u}) = O\left(\frac{T + r^{3/2}}{\sqrt{n}}\right)$$

The entire proof is similar to the previous ones with only few adjustments and can be found in Appendix D.

### 4.5. Multivariate Time Series

Several multivariate versions of DTW and TWED can be found in the literature. Some assume that DTW are computed independently on each dimension, but in this work we focus on the versions given in Definitions 3.1 and 3.4. Based on these definitions, all the theorems proved in the previous sections hold for multivariate times series $X \in \mathbb{R}^{p \times T}$ since the same assumptions hold due to the Lipschitz constants found in Section 3. More details are given in Appendix B.2.

## 5. Training with Gradient Descent

The TWED and DTW are not differentiable because of the use of the "min" function, making it difficult to solve the optimization problem. A differentiable version of the DTW, called soft-DTW, was introduced in (Cuturi & Blondel, 2017). In this section, we introduce a similar version for the TWED, that we call the soft-TWED. It is obtained similarly to soft-DTW, replacing the minimum with the softmin function. Then, to solve the optimization problem $\min_u \hat{R}_n(u)$, we simply replace the distance (DTW or TWED) by its soft version and we perform a gradient descent.

**Soft-DTW.** The Soft Dynamic Time Warping (Soft-DTW) (Cuturi & Blondel, 2017) is a differentiable variant of DTW distance, which provides a smooth approximation to the minimum-cost alignment between two time series. The

gradient can be computed efficiently using a backward dynamic programming recursion detailed in (Cuturi & Blondel, 2017).

**Soft-TWED.** We now propose a version of the soft-TWED measure, based on the same idea as soft-DTW. Similarly to the Soft-DTW, for $\theta \geq 0$, we define the Soft-TWED $\delta_{\lambda,\nu}^{\theta}$ as in Definition 3.4 replacing the $\min$ operator by $\min^{\theta}$ where:

$$\min^{\theta}\{a_1, \ldots, a_l\} := \begin{cases} \min_{1 \leq i \leq l} a_i, & \theta = 0, \\ -\theta \log \sum_{i=1}^{l} e^{-a_i/\theta}, & \theta > 0 . \end{cases} \tag{5}$$

Differentiating algorithmically $\delta_{\lambda,\nu}^{\theta}(X, X')$ requires doing first a forward pass to store all intermediary computations and recover $D^{\theta} = [D_{i,j}^{\theta}]_{\substack{1 \leq i \leq l \\ 1 \leq j \leq m}} = [r_{i,j}]_{\substack{1 \leq i \leq l \\ 1 \leq j \leq m}}$ . Then, the gradient can be efficiently computed using a backward recursion process explained in Appendix E.

For the DTW and the TWED, the computational cost for $N$ time series of size $T$ in dimension $d$, is in $\mathcal{O}(NT^2d)$ per epoch. An illustration is available in Appendix F. Note that alternative versions of DTW restrict the search to a subset of possible temporal alignments, thereby reducing the computational cost (Itakura, 1975; Sakoe & Chiba, 1978). Exploring such variants could be an interesting direction for future work.

# 6. Experiments

In this section, we first give a simple example that motivates the use of TWED/DTW distances rather than $L^2$. Then, we demonstrate the advantages of our method over previous approaches using synthetic data. Finally, we apply our method on real data from the UCR time series archive (Dau et al., 2019). Additional experiments on semi-synthetic data are presented in Appendix G.6

Note that, while our method learns from pairwise comparisons, it can return a ranking where the samples are ranked depending on their distance to the estimated ideal point. Thus, to evaluate our method, we use both the accuracy regarding pairwise comparison and metrics measuring the ranking similarity such as the Kendall's tau (Kendall, 1938). For synthetic data, we also quantify the distance between the ideal point and the estimated one. In all the experiments, we use the Hinge loss and the gradient descent is performed using PyTorch and ADAM optimizer (Kingma & Ba, 2017). Moreover, we take a zero signal for the initial point.

## 6.1. Motivating Example

The key advantage of our method is its robustness to time-warping deformations. To illustrate this, we display in Fig-

ure 1 an example showing that when using the $L^2$ distance, simply shifting or dilating the input signals in time can lead to different distance values and thus, to different rankings. In other words, IPM with the $L^2$ norm is highly sensitive to temporal misalignment. In contrast, DTW is invariant to time shifts and dilations, as it explicitly aligns the signals before computing the distance. As a result, DTW yields consistent distances and stable rankings under such transformations, allowing the method to recover the underlying ideal pattern.

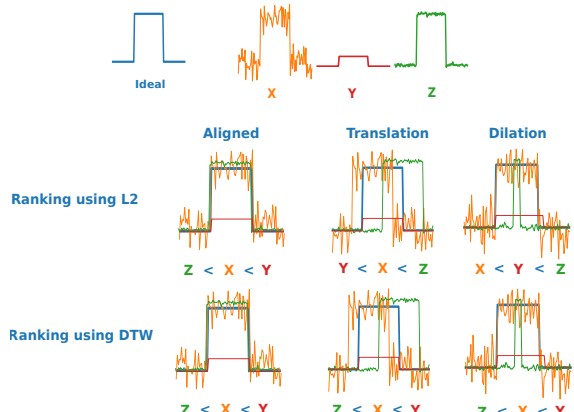

*Figure 1.* Toy example illustrating the differences of ranking between $L^2$ and DTW.

## 6.2. Synthetic Data

In this section, we evaluate the efficiency of our method by first showing that, when data follow the IPM model, the ideal can be retrieved, even in the presence of noise (as defined in 4.8). In addition, on a synthetic dataset where the true ranking is determined by the noise level of the signals, we compare our method to the standard case where $d$ is the $L^2$ distance. A short empirical study to evaluate the effectiveness of the low dimensional model (see Section 4.4) is also presented in Appendix G.4.

### 6.2.1. SOFT APPROXIMATION

In this experiment, synthetic data are generated by performing time warping and adding noise to an ideal time series $u^* \in \mathbb{R}^T$ (a bipolar pulse of length $T = 30$ in our experiments). More specifically, let $\sigma_1, \sigma_2 \sim \mathcal{U}(0, 0.5)$ and let $\varphi_1, \varphi_2$ be two warping functions with a warp strength $w_s = 2$. We generate $X_1 = \varphi_1(u^*) + \varepsilon_1$ and $X_2 = \varphi_2(u^*) + \varepsilon_2$, with $\varepsilon_1 \sim \mathcal{N}(0_{\mathbb{R}^T}, \sigma_1 I_T)$ and $\varepsilon_2 \sim \mathcal{N}(0_{\mathbb{R}^T}, \sigma_2 I_T)$. The preference label $y \in \{-1, 1\}$ is then generated according to Definition 4.8 where $f_\tau : x \mapsto \frac{1}{1+\exp(-x/\tau)}$ and $d$ is either the DTW or the TWED distance. More details regarding this synthetic generation are given in Appendix G.1.

For each value of $n \in \{50, 250, 500, 1000, 2000, 4000\}$, the dataset $S$ is generated taking $n$ triplets $(X_1, X_2, y)$ for both the training set and the test set. We repeat this process

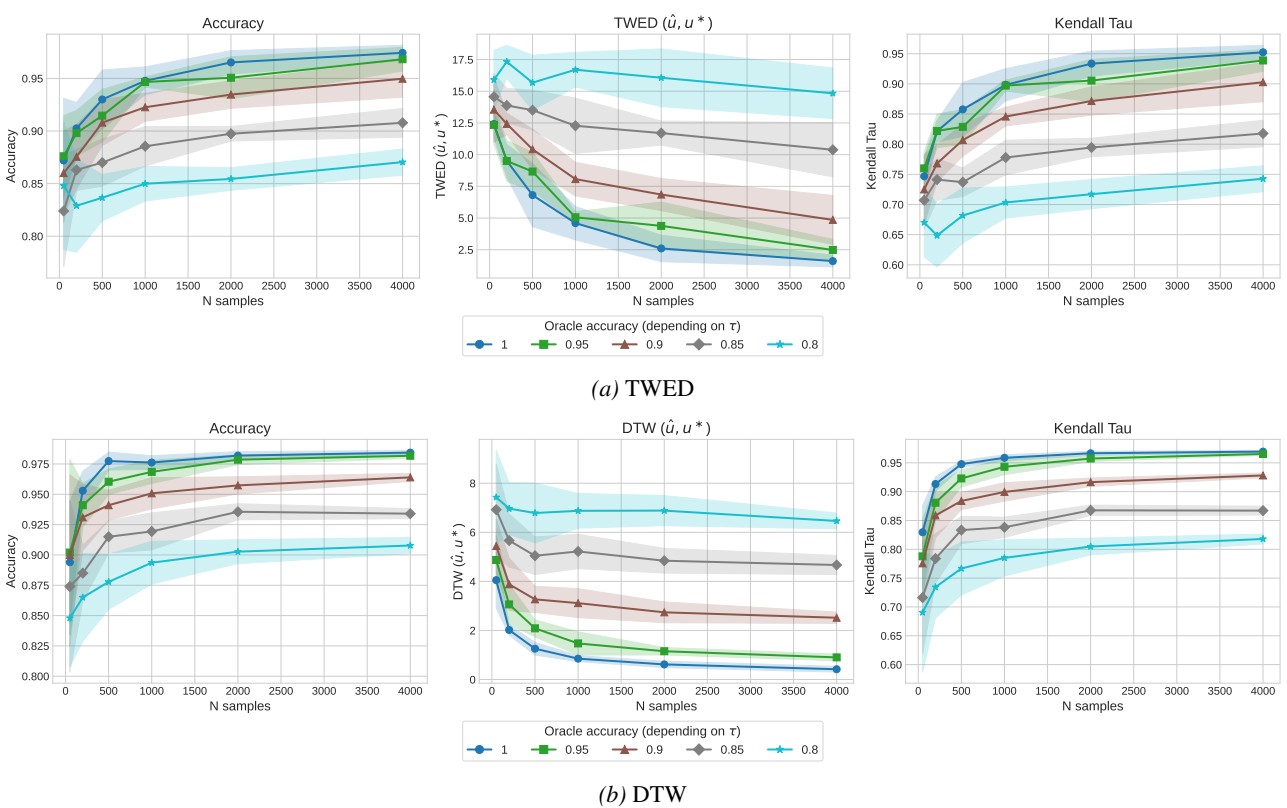

*Figure 2.* Convergence results under several noise levels for the TWED and DTW. The soft-min parameter $\theta$ is set to $0.1$.

10 times to obtain 10 independent datasets. Finally, the optimization was done with a learning rate equal to $0.01$ and with $300$ epochs.

The objective of this experiment is to analyze the convergence of our model under several levels of noise and the impact of the number of samples $n$.

**Results:** Figure 2 illustrates the convergence of our method with the two distances (TWED and DTW), as a function of the number of training points $n$, through three metrics: the accuracy (computing the rate of true preferences predicted), the distance (TWED/DTW) between the estimated ideal and the true one, and the Kendall Tau (Kendall, 1938) to evaluate the final ranking. The different curves indicate the results obtained under different levels of noise (depending on $\tau$): an oracle accuracy of $1$ means a perfect annotation while an oracle accuracy of $0.8$ indicates that $20\%$ of the annotations are wrong.

For the TWED distance $\delta_{\lambda,\nu}$, we set $\lambda = 0.5$ and $\nu = 0.01$. It defines a distance with high elasticity while penalizing the delete operation. In Figure 2a, the three metrics (accuracy, $\delta_{\lambda,\nu}(\hat{u}_n, u^*)$ and Kendall Tau) consistently improve as the number of samples increases regardless of the level of noise. Indeed, the accuracy curves rapidly approach their oracle levels, while Kendall Tau steadily increases, indicat-

ing that the learned ranking progressively aligns with the true ordering. Regarding $\hat{u}_n$, we observe that it converges towards $u^*$ for high oracle accuracy settings. In contrast, when this accuracy is below $0.9$, the convergence is not reached yet, even with $n = 4000$. However, the accuracy of the trained model becomes better than the oracle accuracy in these cases which is a desired property. For the DTW, the same conclusions are valid (see Figure 2b). Note that the parameter $\theta$ of the soft-min can impact the results. An additional analysis on this is provided in Appendix G.2.

In conclusion, this experiment shows that, under reasonable noise conditions, our method provides an accurate estimate of the ideal point $u^*$ and gives robust results in terms of final ranking.

### 6.2.2. RANKING FROM NOISE LEVEL

In this section, time series are generated according to the same model as described in the previous section but preference labels are given by $y = \text{sign}\big(-(\sigma_1 - \sigma_2)\big)$. This means that the ranking is determined by the amount of noise in the signals, reflecting the idea that an annotator always prefers the less noisy version to the noisier one. The main objective is therefore to demonstrate the ability of our models to retrieve the noise ranking (the more noisy, the higher rank)

| Warp | 0 | | | 0.5 | | | 1 | | | 3 | | |
|---|---|---|---|---|---|---|---|---|---|---|---|---|
| $n$ | 50 | 500 | 2000 | 50 | 500 | 2000 | 50 | 500 | 2000 | 50 | 500 | 2000 |
| DTW | 0.65 (0.13) | 0.81 (0.06) | **0.85** (**0.01**) | 0.56 (0.2) | 0.72 (0.05) | 0.76 (0.0) | 0.62 (0.05) | 0.72 (0.04) | **0.76** (**0.01**) | 0.41 (0.15) | **0.69** (**0.05**) | **0.73** (**0.02**) |
| TWED | **0.76** (**0.05**) | **0.82** (**0.01**) | 0.83 (0.0) | **0.71** (**0.04**) | **0.78** (**0.02**) | **0.8** (0.0) | **0.67** (**0.02**) | **0.74** (**0.01**) | 0.76 (0.02) | **0.54** (**0.09**) | 0.67 (0.03) | 0.68 (0.04) |
| $L^2$ | 0.37 (0.08) | 0.75 (0.02) | 0.83 (0.01) | 0.25 (0.04) | 0.48 (0.02) | 0.52 (0.01) | 0.24 (0.06) | 0.42 (0.01) | 0.46 (0.01) | 0.13 (0.08) | 0.23 (0.02) | 0.25 (0.02) |

*Table 1.* Results for the Kendall metric. The higher the better.

even in scenarios with time warping.

The experiments are conducted under the same conditions and with the same parameters as those described in Section 6.2.1. To highlight the advantage of our method when working with time series, we compare it to the IPM with the $L^2$ distance. Here, the labels are noiseless and we compare the results of the three models (DTW, TWED, $L^2$) for varying warp strength parameters.

**Results:** Table 1 clearly shows that, while all methods improve when the sample size $n$ increases, $L^2$ fails when the warp strength increases, exhibiting significantly lower Kendall. In contrast, both DTW and especially TWED remain robust across all warp strengths, maintaining high performance and low variance. TWED consistently achieves the best results, demonstrating superior stability and accuracy even under strong time-series distortions. These findings emphasize the robustness of our methods compared to the vulnerability of $L^2$ to temporal warping. The same conclusions can be made looking at the Spearman rank correlation and the accuracy scores (cf table in Appendix G.3).

## 6.3. Real Data

We now assess the performance of our method on a real-world multivariate time series dataset.

**Dataset description:** We used the Basic Motions dataset from the UCR time series archive (Dau et al., 2019). Data was recorded by a smart watch during four activities. The watch collected 3D accelerometer and a 3D gyroscope. There are 4 classes: walking, resting, running and badminton. The data is sampled at 10Hz during 10 seconds. Given a permutation $\xi$ of $(1, 2, 3, 4)$, we induce an order on the labels by defining $s < t$ if and only if $\xi(s) < \xi(t)$. This gives us a (partial) true ranking. The dataset is then transformed into triplets $(x, x', y)$, where $x$ and $x'$ are two time series with distinct classes $l^1$ and $l^2$, and the preference label is defined as $y = \text{sign}(\xi(l^1) - \xi(l^2))$. All time series are re-normalized between $[-1, 1]$ prior to training. The dataset is split into 40 signals for training and 40 for testing, and model performance is evaluated on the test set using the Kendall Tau metric. The experiments were run for

all the 24 permutations of $(1, 2, 3, 4)$ (corresponding to the 24 possible orderings, i.e possible kinds of preferences).

**Baselines:** In addition to the standard IPM model based on the $L^2$ distance, we included two learning-to-rank baselines: RankNet (Burges et al., 2005) and RankSVM (Joachims, 2002). Since these methods operate on fixed-dimensional feature vectors in $\mathbb{R}^D$, we consider feature-based representations obtained with Catch22 (Lubba et al., 2019) and ROCKET (Dempster et al., 2020), two widely used time-series feature extractors. These representations are then used as inputs to RankNet and RankSVM. The feature-based RankNet model is implemented as a neural network with one hidden layer of 64 units, a ReLU activation, and a linear output layer producing the ranking score. We also consider a convolutional RankNet variant that operates directly on raw time series. This architecture consists of three one-dimensional convolutional layers with 16, 32, and 64 channels, respectively, each using a kernel size of 3 and ReLU activations. The first two convolutional layers are followed by max-pooling layers, and an adaptive average pooling layer is then used to obtain a fixed-dimensional representation. This representation is then passed through a fully connected neural network of a 64-unit hidden layer with ReLU activation and a final linear output layer.

**Results:** Table 2 reports the Kendall's tau scores obtained on the Basic Motions dataset over all 24 possible label permutations. Our method with DTW or TWED, returns strong performance, with average scores of $0.69$ and $0.70$, respectively. They achieve higher performance than the standard IPM model based on the $L^2$ distance (even when we use Catch22), highlighting the benefit of using temporal alignment rather than Euclidean comparisons. They also compare favorably with almost all other baselines. The strongest baseline is RankSVM combined with ROCKET(1000), which gives the best Kendall's tau. Nevertheless, DTW and TWED remain competitive while operating directly on the original time series and, importantly, benefit from the interpretability of the IPM framework. In particular, our method enables the visualization and interpretation of the estimated ideal signal, which is not directly available for the feature-based ranking base-

| Method | Kendall Tau ↑ |
|---|---|
| **DTW** | 0.69 (0.08) |
| **TWED** | 0.70 (0.10) |
| $\boldsymbol{L^2}$ | 0.42 (0.26) |
| **RankNet** | 0.61 (0.10) |
| **Catch22 $L^2$** | $0.27 \pm (0.3)$ |
| **Catch22 RankNet** | $0.61 \pm (0.1)$ |
| **Catch22 RankSVM** | $0.66 \pm (0.1)$ |
| **RankNet CNN** | $0.66 \pm (0.12)$ |
| **RankNet Rocket (200)** | $0.58 \pm (0.08)$ |
| **RankSVM Rocket (200)** | $0.65 \pm (0.07)$ |
| **RankNet Rocket (1000)** | $0.66 \pm (0.1)$ |
| **RankSVM Rocket (1000)** | $0.83 \pm (0.02)$ |

*Table 2.* Kendall Tau scores on the Basic Motions dataset.

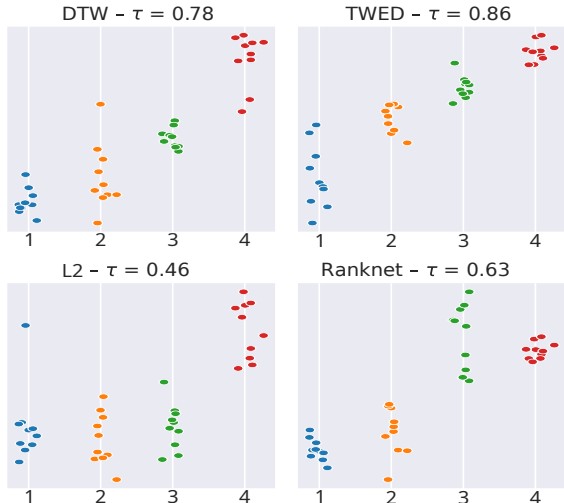

*Figure 3.* Example of distances-to-ideal for each method on Basic Motions. Here, 0 = Walking, 1 = Standing, 2 = Badminton, 3 = Running.

lines. A visualization of the rankings returned by the different methods for one permutation is provided in Figure 3. Additional examples of the estimated ideal signal, together with signals assigned lower and higher ranks, are reported in Appendix G.5.

## 7. Conclusion

We introduced an IPM to rank time series from pairwise comparisons using elastic distances. First, we proved key properties on the DTW and TWED. Building on these results, we established theoretical guarantees for our proposed models. Finally, we proposed an optimization method effectively leading to better results than the standard L2 model, in both univariate and multivariate settings, for both synthetic and real data. The main limitation of our approach lies in the assumption of a single ideal point used to rank the time series. While this simplifies the model, it may not fully capture datasets where multiple ideal patterns

exist. Extending the method to accommodate several ideal points could be a promising direction for future work (see e.g Appendix G.7). Another limitation is the computational cost associated with TWED and DTW for long time series. This is partly mitigated by the low-rank formulation we introduce, which significantly reduces the complexity while preserving ranking performance.

## Impact Statement

This paper presents work whose goal is to advance the field of Machine Learning. There are many potential societal consequences of our work, none which we feel must be specifically highlighted here.

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

# A. TWED and DTW proofs.

## A.1. DTW Lipschitz

**Definition A.1.** Given two 1D series of lengths $l$ and $m$, we define

$$\mathcal{A}_{l,m} \subset \{0,1\}^{l \times m}$$

as the set of *binary alignment matrices*. Each $A \in \mathcal{A}_{l,m}$ represents a path on an $l \times m$ grid that connects the upper-left entry $(1,1)$ to the lower-right entry $(l,m)$, using only moves of type $\downarrow$, $\rightarrow$, or $\searrow$.

Let $X = (x_1, \ldots, x_l) \in \mathbb{R}^l$ and $X' = (x'_1, \ldots, x'_m) \in \mathbb{R}^m$ be two time series. Given the cost matrix

$$\Delta(X, X') := \left(\delta(x_i, x'_j)\right)_{i,j} \in \mathbb{R}^{l \times m},$$

the DTW can be written as

$$\mathrm{DTW}(X, X') := \min_{A \in \mathcal{A}_{l,m}} \langle A, \Delta(X, X') \rangle.$$

*Proof of Proposition 3.3.* **Univariate.**
Let $X = (x_1, ..., x_T)$, $X' = (x'_1, ..., x'_T)$ and $W = (w_1, ..., w_T)$ in $\mathbb{R}^T$ and let $\tilde{X} = X' + W$.
Then $\tilde{\Delta} = (|x_i - x'_j - w_j|)_{1 \le i,j \le T}$. Using the triangular inequality we have

$$\tilde{\Delta} \preceq \Delta + \left(|w_1| \begin{pmatrix} 1 \\ \vdots \\ 1 \end{pmatrix}, ..., |w_T| \begin{pmatrix} 1 \\ \vdots \\ 1 \end{pmatrix}\right) = \Delta + D.$$

Thus, $\forall A \in \mathcal{A}_T$, $\langle A, \tilde{\Delta} \rangle \le \langle A, \Delta \rangle + \langle A, D \rangle$, and for $A^* \in \arg\min_{A \in \mathcal{A}_T} \langle A, \Delta \rangle$, we have

$$\mathrm{DTW}(X, \tilde{X}) \le \langle A^*, \tilde{\Delta} \rangle \le \mathrm{DTW}(X, X') + \langle A^*, D \rangle$$

then,

$$\mathrm{DTW}(X, \tilde{X}) - \mathrm{DTW}(X, X') \le \langle A^*, D \rangle .$$

Now we need to bound $\langle A^*, D \rangle$. First, remark that in the worst case, starting from $(1,1)$ to $(T,T)$, we sum each $|w_i|$ exactly one time. Moreover, if only one diagonal move is used, we add $(T-2)$ times entries of the matrix $D$ (indeed, the longest minimum path is $(1,1) \rightarrow (1, T-1) \rightarrow (2, T) \rightarrow (T, T)$, since a path passing by $(1, T)$ has necessarily a larger cost). Thus, in worst case scenario, we have that

$$\langle A^*, D \rangle = (T-2)||W||_\infty + ||W||_1.$$

Let $f(z) = (T-2)||z||_\infty + ||z||_1$, by homogeneity of the norm, we are looking for $\max_{z \in \mathbb{R}^T, ||z||_2 = 1} f(z)$. First we assume without loss of generality that $|z_1| = ||z||_\infty$, then

$$f(z) = (T-1)|z_1| + \sum_{i=2}^{T} |z_i|$$

$$\le (T-1)|z_1| + \sqrt{T-1}\sqrt{1 - z_1^2} = g(|z_1|) \qquad \text{Cauchy-Schwarz}$$

Computing the derivative, we find that $g$ reaches its maximum in $|z_1| = \sqrt{\frac{T-1}{T}}$, resulting in a maximum equal to $\sqrt{T(T-1)}$.
Then, let $z \in \mathbb{R}^T$, $f(z) = f(||z||_2 \frac{z}{||z||_2}) = ||z||_2 f(\frac{z}{||z||_2}) \le \sqrt{T(T-1)}||z||_2$, thus:

$$\mathrm{DTW}(X, \tilde{X}) - \mathrm{DTW}(X, X') \le \sqrt{T(T-1)}||\tilde{X} - X'||_2.$$

The same demonstration is valid in the other side, so finally:

$$|\mathrm{DTW}(X, \tilde{X}) - \mathrm{DTW}(X, X')| \leq \sqrt{T(T-1)}||X' - \tilde{X}||_2.$$

Note that the bound is reached taking $X = (0, 1, ..., 1)$, $X' = (0, ..., 0, 1)$ and $\tilde{X} = X' + \alpha w$, where $\alpha = \frac{\sqrt{T(T-1)}}{T(T-1)+2}$ and $w = (a_T, ..., a_T, b_T)$ with $b_T = \sqrt{\frac{T-1}{T}}$ and $a_T = \sqrt{\frac{1-b_T^2}{T-1}} = \frac{1}{\sqrt{T(T-1)}}$. Indeed, taking the same notations:

$$\Delta = \begin{pmatrix} 0 & \dots & 0 & 0 & 1 \\ 1 & \dots & 1 & 1 & 0 \\ \vdots & \ddots & \vdots & \vdots & \vdots \\ 1 & \dots & 1 & 1 & 0 \end{pmatrix}, \quad \tilde{\Delta} = \begin{pmatrix} \alpha a_T & \dots & \alpha a_T & \alpha a_T & 1 + \alpha b_T \\ 1 - \alpha a_T & \dots & 1 - \alpha a_T & 1 - \alpha a_T & \alpha b_T \\ \vdots & \ddots & \vdots & \vdots & \vdots \\ 1 - \alpha a_T & \dots & 1 - \alpha a_T & 1 - \alpha a_T & \alpha b_T \end{pmatrix}.$$

Let $\pi$ be the path taking all the 0 costs in $\Delta$. Note that the same path on $\tilde{\Delta}$ has a cost $\alpha\sqrt{T(T-1)} = \frac{T(T-1)}{T(T-1)+2}$. Then, if another path is taken on $\tilde{\Delta}$, the cost will be larger than $1 - \alpha a_T = 1 - \frac{1}{T(T-1)+2} = \frac{T(T-1)+1}{T(T-1)+2}$ which is larger than with path $\pi$. Thus the minimum is reached with the path $\pi$ so $DTW(X, \tilde{X}) = \alpha\sqrt{T(T-1)}$. Then, since $DTW(X, X') = 0$ and $||X' - \tilde{X}||_2 = \alpha||w||_2 = \alpha$, we have that:

$$\left| DTW(X, X') - DTW(X, \tilde{X}) \right| = \sqrt{T(T-1)}||X' - \tilde{X}||_2 .$$

**Multivariate.**
The proof is similar to the univariate case.
Let $X = (x_1, ..., x_T)$, $X' = (x'_1, ..., x'_T)$ and $w = (w_1, ..., w_T)$ in $\mathbb{R}^{p \times T}$ and let $\tilde{X} = X' + w$.
Then $\tilde{\Delta} = (||x_i - x'_j - w_j||_q)_{1 \leq i, j \leq T}$. Using the triangular inequality we have

$$\tilde{\Delta} \preceq \Delta + \left( ||w_1||_q \begin{pmatrix} 1 \\ \vdots \\ 1 \end{pmatrix}, ..., ||w_T||_q \begin{pmatrix} 1 \\ \vdots \\ 1 \end{pmatrix} \right) = \Delta + D.$$

Thus, $\forall A \in \mathcal{A}_n$, $\langle A, \tilde{\Delta} \rangle \leq \langle A, \Delta \rangle + \langle A, D \rangle$, and for $A^* = \arg\min\langle A, \Delta \rangle$, we have

$$\mathrm{DTW}_q(X, \tilde{X}) \leq \langle A^*, \tilde{\Delta} \rangle \leq \mathrm{DTW}_q(X, X') + \langle A^*, D \rangle,$$

then,

$$\mathrm{DTW}_q(X, \tilde{X}) - \mathrm{DTW}_q(X, X') \leq \langle A^*, D \rangle$$

Let $v = (||w_1||_q, ..., ||w_T||_q) \in \mathbb{R}^{p \times T}$, as for the univariate case, we have that:

$$\langle A^*, D \rangle = (T - 2)||v||_\infty + ||v||_1.$$

. Then,

$$\langle A^*, D \rangle \leq \sqrt{T(T-1)}||v||_2$$
$$= \sqrt{T(T-1)} \sqrt{\sum_{i=1}^{T} ||w_i||_q^2}.$$

If $q \geq 2$, $||w_i||_q \leq ||w_i||_2$ so

$$\langle A^*, D \rangle \leq \sqrt{T(T-1)}||w||_2.$$

If $q \in [1, 2[$, the Hölder inequality gives that for all $x \in \mathbb{R}^p$, $||x||_q \leq p^{\frac{1}{q} - \frac{1}{2}}||x||_2$ and this concludes the proof.
Finally, note that like for the univariate case, the bounds are reached. Indeed, taking the same example than the univariate case and, for $q \leq 2$ repeating the signal on each of the $p$ dimensions, for $q \geq 2$ setting the dimensions 2 to $p$ to 0, we reach the Lipschitz bound. $\qquad\square$

### A.2. TWED Lipschitz

**Proposition A.2.** *For $X, X'$ two time series of equal length, if $\delta(x, x') = ||x - x'||_1$, then:*

$$\delta_{\lambda,\nu}(X, X') \leq 2 \cdot ||X - X'||_1 \ .$$

*Proof.* The proof is given in (Marteau, 2009). $\qquad\square$

*Proof of Proposition 3.5.* **Univariate.**
The triangular inequality gives
$$\delta_{\lambda,\nu}(X, X') - \delta_{\lambda,\nu}(X, X'') \leq \delta_{\lambda,\nu}(X', X'').$$

then using Proposition A.2, $\delta_{\lambda,\nu}(X', X'') \leq 2||X' - X''||_1 \leq 2\sqrt{T}||X' - X''||_2$.
Note that the Lipschitz constant can be slightly refined. Indeed, the inequality A.2 can be adjusted:

$$\delta_{\lambda,\nu}(X', X'') \leq 2 \sum_{i=1}^{T-1} |X_i' - X_i''| \ + |X_T' - X_T''|$$

.
Then Cauchy-Schwarz applied to the vectors $(X' - X'')$ and $(2, ..., 2, 1)$ gives:

$$\delta_{\lambda,\nu}(X', X'') \leq \sqrt{4T - 3}||X' - X''||_2.$$

Note that the bound is reached taking for example $X = (1, ..., 1), Y = (1, ..., 1), X'' = (1 + 2\varepsilon, ..., 1 + 2\varepsilon, 1 + \varepsilon)$ with $\varepsilon$ small (depending on $\lambda$ and $\nu$. Indeed, we have $\delta_{\lambda,\nu}(X, X') = 0$, $\delta_{\lambda,\nu}(X, X'') = \varepsilon(4T - 3)$ and $||X' - X''||_2 = \varepsilon\sqrt{4T - 3}$.

**Multivariate.**
The triangular inequality gives
$$\delta_{\lambda,\nu}(X, X') - \delta_{\lambda,\nu}(X, X'') \leq \delta_{\lambda,\nu}(X', X'').$$

Then, $\delta_{\lambda,\nu}(X', X'') \leq 2 \sum_{i=1}^{T} ||X_i' - X_i''||_q \leq 2p^{\max(0, \frac{1}{q} - \frac{1}{2})} \sum_{i=1}^{T} ||X_i' - X_i'||_2 \leq 2p^{\max(0, \frac{1}{q} - \frac{1}{2})}\sqrt{T}||X' - X''||_2$.
As for the univariate case, a better constant is $p^{\max(0, \frac{1}{q} - \frac{1}{2})}\sqrt{4T - 3}$ and it is reached taking for example $X$ and $X'$ the times series equal to $(1, 0, ..., 0)$ for each timestamps and $X'' = X' + \epsilon(2, ..., 2, 1)$. $\qquad\square$

## B. Excess Risk Bounds

### B.1. Univariate Case

Recall the following notations and definitions. Let $u \in \mathbb{R}^T$ be a univariate time series and let $S = ((X_1, X_1', y_1), ..., (X_n, X_n', y_n))$ a dataset of iid tuples with two time series $X, X'$ in $\mathbb{R}^T$ and a label $y \in \{-1, 1\}$. Let $d$ be a similarity measure such that for all $u \in \mathbb{R}^T$, $x \mapsto d(u, x)$ is $L_d$-Lipschitz (here $d$ can be the TWED distance for instance). Let $\ell$ be a loss function (the Soft Margin Rank loss for instance). The quantity $\hat{R}_n(u)$ is the empirical risk, given dataset $S$. We define the empirical risk as:

$$\hat{R}_n(u) := \frac{1}{n} \sum_{i=1}^{n} \ell(y_i(d(X_i, u) - d(X_i', u))) \tag{6}$$

The empirical risk is an unbiased estimate of the true risk given by

$$R(u) := \mathbb{E}\left[\ell(y_i(d(X_i, u) - d(X_i', u)))\right]. \tag{7}$$

where the expectation is with respect to a random draw of $i$. Let $\hat{u}_n$ denote ne minimizer in $E$ of the empirical risk optimization and let $\tilde{u}$ minimizes the true risk in $E$.

We now introduce additional useful results to prove Theorem 4.4.

**Lemma B.1.** *Let $f$ and $g$ be two functions from $X$ to $\mathbb{R}$. Then $|\sup_x f(x) - \sup_x g(x)| \leq \sup_x |f(x) - g(x)|$*

*Lemma* (B.1). $f = f - g + g$ so we can write:

$$\sup f \leq \sup(f - g) + \sup g$$
$$\leq \sup|f - g| + \sup g$$

thus $\sup f - \sup g \leq \sup|f - g|$. Doing the same with $g = g - f + f$, we conclude the proof. □

**Lemma B.2.** *(Wainwright, 2019) Let $\mathcal{X}$ a bounded set in $\mathbb{R}^n$, $\sigma_1, ..., \sigma_n$ iid random variables following a Rademacher distribution ($\mathbb{P}(\sigma_1 = -1) = \mathbb{P}(\sigma_1 = 1) = \frac{1}{2}$) and let $g_1, ..., g_n$ iid random variables following $\mathcal{N}(0, 1)$. Then*

$$\mathbb{E}_\sigma \left[ \sup_{x \in \mathcal{X}} \frac{1}{n} \sum_{i=1}^n \sigma_i x_i \right] \leq \sqrt{\frac{\pi}{2}} \mathbb{E}_g \left[ \sup_{x \in \mathcal{X}} \frac{1}{n} \sum_{i=1}^n g_i x_i \right].$$

*Lemma* (B.2).

$$\mathbb{E}_\sigma \left[ \sup_{x \in \mathcal{X}} \frac{1}{n} \sum_{i=1}^n \sigma_i x_i \right] = \mathbb{E}_g \left[ \sup_{x \in \mathcal{X}} \frac{1}{n} \sum_{i=1}^n sign(g_i) x_i \right]$$

$$= \mathbb{E}_g \left[ \sup_{x \in \mathcal{X}} \frac{1}{n} \sum_{i=1}^n sign(g_i) \mathbb{E}[|u_i|] \sqrt{\frac{\pi}{2}} x_i \right] \qquad \text{where } u_i \sim \mathcal{N}(0, 1)$$

For any $x = (x_1, ..., x_n) \in \mathcal{X}$, $(g_1, ...g_n) \in \mathbb{R}^n$, the function $z \mapsto \frac{1}{n} \sum_{i=1}^n sign(g_i) z_i \sqrt{\frac{\pi}{2}} x_i$ is convex (as a linear function), so as the supremum wrt to $\mathcal{X}$. Then using the Jensen inequality,

$$\mathbb{E}_g \left[ \sup_{x \in \mathcal{X}} \frac{1}{n} \sum_{i=1}^n sign(g_i) \mathbb{E}[|u_i|] \sqrt{\frac{\pi}{2}} x_i \right] \leq \sqrt{\frac{\pi}{2}} \mathbb{E}_{g,u} \left[ \sup_{x \in \mathcal{X}} \frac{1}{n} \sum_{i=1}^n sign(g_i) |u_i| x_i \right] = \sqrt{\frac{\pi}{2}} \mathbb{E}_g \left[ \sup_{x \in \mathcal{X}} \frac{1}{n} \sum_{i=1}^n g_i x_i \right].$$

□

**Lemma B.3.** *Let $d_1$ and $d_2$ be two distances over the space $\mathbb{R}^d$ such that for all $x \in \mathbb{R}^d$,*

$$d_1(x, .) \leq C d_2(x, .) \quad (resp\ d_1(x, .) = C d_2(x, .) ),$$

*for some $C > 0$. Let $N(E, d, \varepsilon)$ be the minimum necessary number of balls with radius $\varepsilon$ associated to the distance $d$ to cover the space $E$. Then, for all $\varepsilon > 0$,*

$$N(E, d_1, \varepsilon) \leq N(E, d_2, \frac{\varepsilon}{C}) \quad (resp\ N(E, d_1, \varepsilon) = N(E, d_2, \frac{\varepsilon}{C}) ). \tag{8}$$

*Lemma* (B.3). The proof results from the fact that for all $x \in \mathbb{R}^d$, $B_{d_2}(x, \frac{\varepsilon}{C}) \subseteq B_{d_1}(x, \varepsilon)$. □

**Lemma B.4.** *Let $\mathcal{X} \subseteq \mathbb{R}^d$ such that for all $x \in \mathcal{X}$, $||x||_2 \leq M$, then $N(\mathcal{X}, ||.||_2, \varepsilon) \leq \left( \frac{3M}{\varepsilon} \right)^d$ for $\epsilon \leq M$, and $N(\mathcal{X}, ||.||_2, \varepsilon) = 1$ otherwise.*

*Proof.* http://www.stat.yale.edu/~yw562/teaching/598/lec14.pdf give that:

$$N(\mathcal{X}, ||.||_2, \epsilon) \leq \left( \frac{3}{\epsilon} \right)^d \frac{Vol(X)}{Vol(B)},$$

where $B$ is the unit ball associated to $||.||_2$. Then $X \subseteq B_{||.||_2}(0, M)$ so

$$N(\mathcal{X}, ||.||_2, \epsilon) \leq \left( \frac{3}{\epsilon} \right)^d \frac{Vol(B_{||.||_2}(0, M))}{Vol(B)}.$$

Then, using the change of variable $\phi : x \mapsto Mx$, we have that

$$Vol(B_{||.||_2}(0, M)) = M^d Vol(B).$$

□

**Lemma B.5.** *Let $X_1$, $X_2$ two random variables such that $\mathbb{P}(X_1 > a_1) \leq \delta_1$ and $\mathbb{P}(X_2 > a_2) \leq \delta_2$, then*

$$\mathbb{P}(X_1 + X_2 > a_1 + a_2) \leq \delta_1 + \delta_2 \tag{9}$$

*Proof.* The event $\{X_1 + X_2 \geq a_1 + a_2\}$ is included in $\{X_1 \geq a_1\} \cup \{X_2 \geq a_2\}$. Hence, $\mathbb{P}(X_1 + X_2 \geq a_1 + a_2) \leq \mathbb{P}(\{X_1 > a_1\} \cup \{X_2 > a_2\}) \leq \mathbb{P}(X_1 > a_1) + \mathbb{P}(X_2 > a_2) \leq \delta_1 + \delta_2.$ □

*Proof of Theorem 4.4.* This theorem bounds the excess risk of the empirical optimum $R(\hat{u}_n)$ relative to the optimal true risk $R(\tilde{u})$. We first decompose the excess risk as follows:

$$R(\hat{u}_n) - R(\tilde{u}) = R(\hat{u}_n) - \hat{R}_n(\hat{u}_n) + \hat{R}_n(\hat{u}_n) - \hat{R}_n(\tilde{u}) + \hat{R}_n(\tilde{u}) - R(\tilde{u})$$

$$\leq R(\hat{u}_n) - \hat{R}_n(\hat{u}_n) + \hat{R}_n(\tilde{u}) - R(\tilde{u}) \tag{10}$$

$$\leq \sup_{u \in E} \left( R(u) - \hat{R}_n(u) \right) + \left( \hat{R}_n(\tilde{u}) - R(\tilde{u}) \right) \tag{11}$$

First, using Hoeffding inequality, we have

$$\mathbb{P}\left( \hat{R}_n(\tilde{u}) - R(\tilde{u}) \leq \sqrt{\frac{(l_0 + L_d L \gamma)^2 \log(1/\delta)}{2n}} \right) \geq 1 - \delta$$

using that:

$$0 \leq \ell(y_i(d(X_i, u) - d(X_i', u))) \leq L|y_i(d(X_i, u) - d(X_i', u))| + \ell(0) \qquad \ell \text{ is } L\text{-Lipschitz}$$

$$\leq L_d L \gamma + \ell(0)$$

where $\gamma = \sup_{x,x' \in E} ||x - x'||_2$ is finite by assumption and $l_0 := \ell(0)$. Then, we bound the first term applying the McDiarmid inequality to the function $F = \sup_u \left( R(u) - \hat{R}_n(u) \right)$ defined as $F : \mathcal{X}_1 \times ... \times \mathcal{X}_n \to \mathbb{R}$ where $\mathcal{X}_i = \mathcal{X} = E \times E \times \{-1, 1\}$.

Let $z_1, ..., z_n \in \mathcal{X}^n$ where $z_i = (X_i, X_i', y_i)$ and let $\tilde{z}_i = (\tilde{X}_i, \tilde{X}_i', \tilde{y}_i) \in \mathcal{X}$,

$$|F(z_1, z_2, ..., z_i, ..., z_n) - F(z_1, z_2, ..., \tilde{z}_i, ..., z_n)| =$$

$$\left| \sup_u \left( R(u) - \frac{1}{n} \sum_{j=1}^n \ell\big(y_j(d(X_j, u) - d(X_j', u))\big) \right) \right.$$

$$\left. - \sup_u \left( R(u) - \frac{1}{n} \sum_{j=1}^n \ell\big(y_j(d(X_j, u) - d(X_j', u))\big) - \frac{1}{n}\ell\big(\tilde{y}_i(d(\tilde{X}_i, u) - d(\tilde{X}_i', u))\big) + \frac{1}{n}\ell(y_i(d(X_i, u) - d(X_i', u))) \right) \right|$$

$$\leq \frac{1}{n} \sup_u |\ell\big(\tilde{y}_i(d(\tilde{X}_i, u) - d(\tilde{X}_i', u))\big) - \ell(y_i(d(X_i, u) - d(X_i', u)))| \qquad \text{using Lemma (B.1)}$$

$$\leq \frac{1}{n} L \sup_u |\tilde{y}_i(d(\tilde{X}_i, u) - d(\tilde{X}_i', u)) - y_i(d(X_i, u) - d(X_i', u))| \qquad \ell \text{ is } L\text{-Lipschitz}$$

$$\leq \frac{1}{n} L \left[ \sup_u |\tilde{y}_i(d(\tilde{X}_i, u) - d(\tilde{X}_i', u))| + \sup_u |y_i(d(X_i, u) - d(X_i', u))| \right]$$

$$\leq 2 L_d L \gamma / n .$$

Taking the sup wrt $\tilde{z}_i$, we have:

$$\sup_{\tilde{z}_i} |F(z_1, z_2, ..., z_i, ..., z_n) - F(z_1, z_2, ..., \tilde{z}_i, ..., z_n)| \leq 2 L_d L \gamma / n.$$

The McDiarmid inequality gives:

$$\mathbb{P}\left( \sup_u \left( R(u) - \hat{R}_n(u) \right) \leq \mathbb{E}\left[ \sup_u \left( R(u) - \hat{R}_n(u) \right) \right] + \sqrt{\frac{2 L_d^2 L^2 \gamma^2 \log(1/\delta)}{n}} \right) \geq 1 - \delta \tag{12}$$

Now we need to bound $\mathbb{E}\left[\sup_u \left(R(u) - \hat{R}_n(u)\right)\right]$. Let $\varepsilon_1, ... \varepsilon_n$ be iid Rademacher random variables, using the symmetrization lemma and the contraction lemma ($\ell$ is L-Lipschitz), we get:

$$
\begin{aligned}
\mathbb{E}\left[\sup_u \left(R(u) - \hat{R}_n(u)\right)\right] &\leq 2\mathbb{E}\left[\sup_u \left(\frac{1}{n}\sum_{i=1}^{n} \varepsilon_i \ell(y_i(d(X_i, u) - d(X_i', u)))\right)\right] && \text{(symmetrization)} \\
&\leq 2L\mathbb{E}\left[\sup_u \left(\frac{1}{n}\sum_{i=1}^{n} \varepsilon_i y_i(d(X_i, u) - d(X_i', u))\right)\right] && \text{(contraction)} \\
&= 2L\mathbb{E}\left[\sup_u \left(\frac{1}{n}\sum_{i=1}^{n} \varepsilon_i(d(X_i, u) - d(X_i', u))\right)\right] && (\varepsilon_i y_i \text{ has the same distribution than } \varepsilon_i) \\
&\leq 2L\mathbb{E}\left[\sup_u \left(\frac{1}{n}\sum_{i=1}^{n} \varepsilon_i(d(X_i, u))\right)\right] + 2L\mathbb{E}\left[\sup_u \left(\frac{1}{n}\sum_{i=1}^{n} (-\varepsilon_i)(d(X_i', u))\right)\right] \\
&= 4L\mathbb{E}\left[\sup_u \left(\frac{1}{n}\sum_{i=1}^{n} \varepsilon_i d(X_i, u)\right)\right] && (X_i \text{ has the same distribution than } X_i') \\
&\leq 4L\sqrt{\frac{\pi}{2}}\mathbb{E}\left[\sup_u \left(\frac{1}{n}\sum_{i=1}^{n} g_i d(X_i, u)\right)\right] && \text{where } g_i \sim \mathcal{N}(0, 1) \text{ are iid (Lemma (B.2)).}
\end{aligned}
$$

For any $u, x_1, \ldots, x_n \in E$, let $G_u := \frac{1}{n}\sum_{i=1}^{n} g_i d(x_i, u)$ and $f_u : x \mapsto d(x, u)$. $(G_u)_{u \in E}$ is a gaussian process (because $g_i$ are iid gaussian variables). We define the pseudo-metric $\rho$ by:

$$
\rho(u, v) := \sqrt{\mathbb{E}\left[(G_u - G_v)^2\right]} = \frac{1}{n}\sqrt{\sum_{i=1}^{n}(f_u(x_i) - f_v(x_i))^2} = \frac{d_X(f_u, f_v)}{\sqrt{n}} \tag{13}
$$

where $d_X(f_u, f_v) := \sqrt{\frac{1}{n}\sum_{i=1}^{n}(f_u(x_i) - f_v(x_i))^2}$.

Note that $d_X(f_u, f_v) \leq L_d\|u - v\|_2$ since $d$ is $L_d$-Lipschitz.

Now, using Theorem 11.17 in [Ledoux, 1991], we get that:

$$
\mathbb{E}\left[\sup_u \left(\frac{1}{n}\sum_{i=1}^{n} g_i f_u(X_i)\right)\right] \leq 24\int_0^\infty \sqrt{\log N(E, \rho, \varepsilon)}d\varepsilon, \tag{14}
$$

where $N(E, \rho, \varepsilon)$ is the minimum necessary number of balls with radius $\varepsilon$ associated to the distance $\rho$ to cover the space $E$.

Then,

$$
\begin{aligned}
\int_0^\infty \sqrt{\log N(E, \rho, \varepsilon)}d\varepsilon &= \int_0^\infty \sqrt{\log N(E, d_X, \sqrt{n}\varepsilon)}d\varepsilon && \text{(Lemma (B.3))} \\
&= \frac{1}{\sqrt{n}}\int_0^\infty \sqrt{\log N(E, d_X, t)}dt \\
&\leq \frac{1}{\sqrt{n}}\int_0^\infty \sqrt{\log N(E, \|.\|_2, (t/L_d))}dt && \text{(Lemma (B.3))} \\
&\leq \frac{1}{\sqrt{n}}\int_0^\infty \sqrt{\log\left\lceil\frac{3L_d\gamma}{t}\right\rceil^T}dt && \text{(Lemma (B.4))} \\
&= 3L_d\gamma\sqrt{\frac{T}{n}}\int_0^1 \sqrt{\log\left\lceil\frac{1}{u}\right\rceil}du \; ; .
\end{aligned}
$$

$\square$

Since $u \mapsto \sqrt{\log \lceil \frac{1}{u} \rceil}$ is integrable in $0^+$, we have that:

$$\int_0^\infty \sqrt{\log N(E, \rho, \varepsilon)} d\varepsilon \leq 3 L_d C \gamma \sqrt{\frac{T}{n}} \tag{15}$$

where $C = \int_0^1 \sqrt{\log \lceil \frac{1}{u} \rceil} du \simeq 1.14$.
Then,

$$\mathbb{E}\left[\sup_u \left(R(u) - \hat{R}_n(u)\right)\right] \leq \tilde{C} L_d L \gamma \sqrt{\frac{T}{n}} \tag{16}$$

Finally, applying Lemma (B.5), we have that with probability at least $1 - 2\delta$,

$$R(\hat{u}_n) - R(\tilde{u}) \leq \sqrt{\frac{(l_0 + L_d L \gamma)^2 \log(1/\delta)}{2n}} + \sqrt{\frac{2 L_d^2 L^2 \gamma^2 \log(1/\delta)}{n}} + \tilde{C} L_d L \gamma \sqrt{\frac{T}{n}}$$

i.e

$$R(\hat{u}_n) - R(\tilde{u}) \leq (l_0 + 3 L_d L \gamma) \sqrt{\frac{\log(1/\delta)}{2n}} + \tilde{C} L_d L \gamma \sqrt{\frac{T}{n}} . \tag{17}$$

### B.2. Multivariate Case

Note that all the proofs of the previous section still work in the multivariate case when the multivariate DTW and TWED follow Definitions 3.1 and 3.4. The two modifications are the Lipschitz constants that are not the same than in the univariate case and the end of the proof when the covering number is computed. Indeed, the dimension of the space becomes $p \times T$ rather than $T$, leading to:

$$R(\hat{u}_n) - R(\tilde{u}) \leq (l_0 + 3 L_d L \gamma) \sqrt{\frac{\log(1/\delta)}{2n}} + \tilde{C} L_d L \gamma \sqrt{\frac{pT}{n}} . \tag{18}$$

The main difference with the proof of Theorem 4.4 is that, in the multivariate case, we also have $d_X(u, v) \leq L_d \|u - v\|_2$ but now $\|u - v\|_2$ is a matrix-norm. Thus, applying Lemma (B.4) gives that:

$$\frac{1}{\sqrt{n}} \int_0^\infty \sqrt{\log N(E, \|.\|_2, (t/L_d))} \leq \frac{1}{\sqrt{n}} \int_0^\infty \sqrt{\log \left\lceil \frac{3 L_d \gamma}{t} \right\rceil^{pT}} dt .$$

## C. Recovery Guarantees

*Proof of Lemma 4.10.* For any $u$, we have that:

$$R(u) - R(u^*) = \mathbb{E}\left[\mathrm{KL}\big(f\big(-\Delta_{X,X'}(u^*)\big) \,\|\, f\big(-\Delta_{X,X'}(u)\big)\big)\right] .$$

Since the KL is a positive function, we have that $u^*$ is a minimizer of $R$. Furthermore, $R(u) - R(u^*) = 0$ iff $f\big(-\Delta_{X,X'}(u^*)\big) = f\big(-\Delta_{X,X'}(u)\big)$ a.s. Since $f$ is monotonous, we have that $R(u) = R(u^*)$ iff $\Delta_{X,X'}(u^*) = \Delta_{X,X'}(u)$ a.s. The assumption 4.9 implies that $R(u) - R(u^*) = 0$ iif $u = u^*$. In other words, the minimizer of $R$ is unique and is equals to $u^*$. □

*Proof of Theorem 4.11.* First, by Lemma 4.10, $u^* = \tilde{u} := \arg\min R(u)$. Now, let us define $W_n = R(\hat{u}_n) - R(u^*)$. By Theorem 4.4, we have that:

$$\forall \varepsilon > 0, \ \mathbb{P}\left(W_n \geq \varepsilon\right) \leq \exp\left(-\frac{(\varepsilon \sqrt{n} - B)^2}{A}\right),$$

where $A = 12 T L^2 \gamma^2$ and $B = 2 \tilde{C} T L \gamma$. Thus, for all $\varepsilon > 0$, $\sum_n \mathbb{P}\left(W_n \geq \varepsilon\right) < \infty$, and using the corollary of the Borel-Cantelli theorem, since $W_n$ is a non-negative random variable, we have that $W_n \xrightarrow{a.s} 0$. Now, let

$\omega \in \{\omega \in \Omega \mid W_n(\omega) \longrightarrow 0\}$, for all $n \in \mathbb{N}$, $\hat{u}_n(\omega) \in E$ the set of time series which is closed and bounded by assumption, so it is compact. Then, let $v$ an accumulation point of $\hat{u}_n(\omega)$, i.e $\exists \varphi : \mathbb{N} \to \mathbb{N}$ increasing such that $\hat{u}_{\varphi(n)}(\omega) \underset{n}{\longrightarrow} v$. By dominated convergence we have that $R(\hat{u}_{\varphi(n)}(\omega)) - R(u^*) \underset{n}{\longrightarrow} R(v) - R(u^*) = 0$ by definition of $\omega$. Then, by uniqueness of $u^*$, we have that $v = u^*$, so $\hat{u}_n(\omega)$ has a unique accumulation point in a compact so it converges towards $u^*$. Finally, using that $\mathbb{P}\left(\{\omega \in \Omega \mid W_n(\omega) \longrightarrow 0\}\right) = 1$ concludes the proof. $\qquad\square$

# D. Low Dimension Model

## D.1. Lipschitz Constants

Note that in this case of low dimension modeling, two Lipschitz constants appear, $L_d^r$, the one of the function $d_r : \mathbb{R}^T \to \mathbb{R}$ defined by $x \mapsto d(u_r, x)$ and $L_d^T$, the one of $d_T : \mathbb{R}^r \to \mathbb{R}$ defined by $x \mapsto d(u_T, x)$ where $u_r \in \mathbb{R}^r$ and $u_T \in \mathbb{R}^T$. Indeed, these two constants are used in the demonstration of the excess risk bounds (note that there are the same for the case $T = r$).

$L_{\text{TWED}}^r$  Without any additional assumption, taking $u \in \mathbb{R}^r$ and considering $f_u : \mathbb{R}^T \to \mathbb{R}$ defined by $f_u(x) = \delta_{\lambda,\nu}(u, x)$, the Lipschitz constant remains in $O(\sqrt{T})$. For example, let $u = (0, ..., 0) \in \mathbb{R}^r$, $y_1 = (0, ..., 0) \in \mathbb{R}^T$ and $y_2 = (-M, M, -M, M...) \in \mathbb{R}^T$. First, $\delta_{\lambda,\nu}(u, y_1) = (T - r)(\lambda + \nu)$ ($r$ matches with a cost 0 and $T - r$ deletions with a cost $(\lambda + \nu)$). Then, for the computation of $\delta_{\lambda,\nu}(u, y_2)$,the costs are: $2 * M$ for a match (except the first one equal to $M$), $(\lambda + \nu)$ for a deletion in $u$ and $(2M + \lambda + \nu)$ for a deletion in $y_2$. Taking $M$ small enough, the cost is lower considering matching rather than deletion in $U$ ($2M \leq \lambda + \nu$). Thus the final cost is equal to $(2r - 1)M + (2M + \lambda + \nu)(T - r) = (2T - 1)M + (T - r)(\lambda + \nu)$. Then, $||y_1 - y_2||_2 = \sqrt{T}M$, so

$$|\delta_{\lambda,\nu}(u, y_1) - \delta_{\lambda,\nu}(u, y_2)| = \frac{2T - 1}{\sqrt{T}}||y_1 - y_2||_2 .$$

$L_{\text{TWED}}^T$  The low dimension model lead to a refined constant here.

**Proposition D.1.** *Assuming that $\delta(x, y) = ||x - y||_q$, for $X \in \mathbb{R}^{p \times T}$ and $Y, Y' \in \mathbb{R}^{p \times r}$ the TWED distance satisfies:*

$$|\delta_{\lambda,\nu}(X, Y) - \delta_{\lambda,\nu}(X, Y')|$$
$$\leq 2\sqrt{r}\, p^{\max(0, \frac{1}{q} - \frac{1}{2})}||Y - Y'||_2$$

*Proof.* Same proof than standard case. $\qquad\square$

$L_{\text{DTW}}^r$  The low dimension case lead to a refined Lipschitz constant for the DTW.

**Proposition D.2.** *Assuming that $\delta(x, y) = ||x - y||_q$, for $X \in \mathbb{R}^{p \times r}$, and $Y, Y' \in \mathbb{R}^{p \times T}$, the $DTW_q$ distance satisfies:*

$$|DTW(X, Y) - DTW(X, Y')|$$
$$\leq \sqrt{(r - 1)^2 + (T - 1)}\, p^{\max(0, \frac{1}{q} - \frac{1}{2})}||Y - Y'||_2.$$

*Proof.* The proof is exactly the same than for the standard case, replacing

$$\langle A^*, D \rangle = (T - 2)||v||_\infty + ||v||_1$$

by

$$\langle A^*, D \rangle = (r - 2)||v||_\infty + ||v||_1.$$

$\qquad\square$

*Remark* D.3. Note that $\sqrt{(r - 1)^2 + (T - 1)} = O(r + \sqrt{T})$, so we can reduce the Lipschitz constant to $O(\sqrt{T})$, which is the same order than the Lipschitz constant of the TWED.

$L_{\mathbf{DTW}}^T$    The low case modeling results in a slightly better constant, in the same order than the standard case.

**Proposition D.4.** *Assuming that $\delta(x, y) = ||x - y||_q$, for $X \in \mathbb{R}^{p \times T}$, and $Y, Y' \in \mathbb{R}^{p \times r}$, the $DTW_q$ distance satisfies:*

$$|DTW(X, Y) - DTW(X, Y')|$$
$$\leq \frac{(T-1)^2 + (T-1)\sqrt{r-1}}{\sqrt{(T-1)^2 + r - 1}} \, p^{\max(0, \frac{1}{q} - \frac{1}{2})}||Y - Y'||_2.$$

*Remark* D.5.  Note that $\frac{(T-1)^2 + (T-1)\sqrt{r-1}}{\sqrt{(T-1)^2 + r - 1}} = O(T)$ since $r < T$.

### D.2. Excess Risk Bounds

**Theorem D.6.** *Let $S = ((x_1, x'_1, y_1), ..., (x_n, x'_n, y_n))$ a dataset of tuples with two time series $x_1$, $x_2$ in $\mathbb{R}^T$ and a label $y \in \{-1, 1\}$ and let $\hat{u}_n \in \mathbb{R}^r$ be the minimizer of $\hat{R}_n$. We have that with probability at least $1 - 2\delta$,*

$$R(\hat{u}_n) - R(u^*) \leq (l_0 + 3L_d^r L \gamma_T)\sqrt{\frac{\log(1/\delta)}{2n}} + \tilde{C}L_d^T L \gamma_r \sqrt{\frac{r}{n}} \tag{19}$$

*where $\gamma_T = \sup\limits_{x,y \in \mathbb{R}^T} ||x - y||_2$ and $\gamma_r = \sup\limits_{u,v \in \mathbb{R}^r} ||u - v||_2$ are the diameters of the time series set wrt to the L2 norm and $L_d^r$ and $L_d^T$ are the Lipschitz constants defined before.*

*Proof.*  The proof is the same than the one in B.1, replacing $L_d$ and $\gamma$ by the new constants. □

**Corollary D.7.** *In the univariate case, we have that:*

$$\textbf{\textit{TWED}}: \qquad R(\hat{u}_n) - R(u^*) = O\left(\frac{T + r^{3/2}}{\sqrt{n}}\right)$$

$$\textbf{\textit{DTW}}: \qquad R(\hat{u}_n) - R(u^*) = O\left(\frac{Tr}{\sqrt{n}}\right)$$

## E. Soft-TWED Gradient Computation

In this section, we computes the gradients for the general definition of the TWED with general timestamps (Marteau, 2009).

Differentiating algorithmically $\delta_{\lambda,\nu}^\theta$ requires doing first a forward pass to store all intermediary computations and recover $D^\theta = [D_{i,j}^\theta]_{\substack{1 \leq i \leq l \\ 1 \leq j \leq m}} = [r_{i,j}]_{\substack{1 \leq i \leq l \\ 1 \leq j \leq m}}$. The value of $\delta_{\lambda,\nu}^\theta(x, x')$, stored in $r_{l,m}$ at the end of the forward recursion, is then impacted by a change in $r_{i,j}$ exclusively through the terms in which $r_{i,j}$ plays a role, namely the triplet of terms $r_{i+1,j}$, $r_{i,j+1}$, $r_{i+1,j+1}$. A straightforward application of the chain rule then gives

$$\frac{\partial r_{l,m}}{\partial r_{i,j}} = \underbrace{\frac{\partial r_{l,m}}{\partial r_{i+1,j}}}_{e_{i,j}} \underbrace{\frac{\partial r_{i+1,j}}{\partial r_{i,j}}}_{e_{i+1,j}} + \underbrace{\frac{\partial r_{l,m}}{\partial r_{i,j+1}}}_{e_{i,j+1}} \underbrace{\frac{\partial r_{i,j+1}}{\partial r_{i,j}}}_{} + \underbrace{\frac{\partial r_{l,m}}{\partial r_{i+1,j+1}}}_{e_{i+1,j+1}} \underbrace{\frac{\partial r_{i+1,j+1}}{\partial r_{i,j}}}_{},$$

in which we have defined the notation of the main object of interest of the backward recursion:

$$e_{i,j} := \frac{\partial r_{l,m}}{\partial r_{i,j}}.$$

Let $\Delta^{xx'} = [\delta(x_i, x'_j)]$, $\Delta^x = [\delta(x_i, x_j)]$ and $\Delta^{x'} = [\delta(x'_i, x'_j)]$ the cost matrices. The recursion evaluated at $(i + 1, j)$ yields:

$$r_{i+1,j} = \min^\theta \{r_{i,j-1} + \Delta_{i,j-1}^{xx'} + \Delta_{i+1,j}^{xx'} + \nu(|t_{x_{i+1}} - t_{x'_j}| + |t_{x_i} - t_{x'_{j-1}}|),$$
$$r_{i,j} + \Delta_{i,i+1}^x + \lambda + \nu|t_{x_{i+1}} - t_{x_i}|, r_{i+1,j-1} + \Delta_{j,j-1}^{x'} + \lambda + \nu|t_{x'_j} - t_{x'_{j-1}}|\},$$

i.e $r_{i+1,j} = \min^\theta(f_{i,j}, g_{i,j}, h_{i,j})$, which, when differentiated with respect to $r_{i,j}$ yields the ratio:

$$\frac{\partial r_{i+1,j}}{\partial r_{i,j}} = \frac{e^{-g_{i,j}/\theta}}{e^{-f_{i,j}/\theta} + e^{-g_{i,j}/\theta} + e^{-h_{i,j}/\theta}}.$$

The logarithm of that derivative can be conveniently cast using evaluations of $\min^\theta$ computed in the forward loop:

$$\theta \cdot \log \frac{\partial r_{i+1,j}}{\partial r_{i,j}} = \min^\theta\{f_{i,j}, g_{i,j}, h_{i,j}\} - r_{i,j} - \Delta^x_{i,i+1} - \lambda - \nu|t_{x_{i+1}} - t_{x_i}|$$

$$= r_{i+1,j} - r_{i,j} - \Delta^x_{i,i+1} - \lambda - \nu|t_{x_{i+1}} - t_{x_i}|.$$

Similarly, the following relationships can also be obtained:

$$\theta \cdot \log \frac{\partial r_{i,j+1}}{\partial r_{i,j}} = r_{i,j+1} - r_{i,j} - \Delta^{x'}_{j+1,j} - \lambda - \nu|t_{x'_{j+1}} - t_{x'_j}|,$$

$$\theta \cdot \log \frac{\partial r_{i+1,j+1}}{\partial r_{i,j}} = r_{i+1,j+1} - r_{i,j} - \Delta^{xx'}_{i,j} - \Delta^{xx'}_{i+1,j+1} - \nu(|t_{x_{i+1}} - t_{x'_j+1}| + |t_{x_i} - t_{x'_j}|).$$

We have therefore obtained a backward recursion to compute the entire matrix $E = [e_{i,j}]$, starting from

$$e_{n,m} = \frac{\partial r_{l,m}}{\partial r_{l,m}} = 1$$

down to $e_{1,1}$.

Then we need to compute the partial derivatives with respect to $\Delta^{xx'}_{i,j}$ and $\Delta^x_{i+1,i}$. We have that:

$$\frac{\partial r_{l,m}}{\partial \Delta^x_{i+1,i}} = \sum_j \frac{\partial r_{n,m}}{\partial r_{i+1,j}} \frac{\partial r_{i+1,j}}{\partial \Delta^x_{i+1,i}} = \sum_j e_{i+1,j} \frac{\partial r_{i+1,j}}{\partial r_{i,j}}$$

$$\frac{\partial r_{l,m}}{\partial \Delta^{xx'}_{i,j}} = \frac{\partial r_{l,m}}{\partial r_{i,j}} \frac{\partial r_{i,j}}{\partial r_{i-1,j-1}} + \frac{\partial r_{l,m}}{\partial r_{i+1,j+1}} \frac{\partial r_{i+1,j+1}}{\partial r_{i,j}}$$

Finally,

$$\frac{\partial r_{l,m}}{\partial x_i} = \frac{\partial r_{l,m}}{\partial \Delta^x_{i+1,i}} \frac{\partial \Delta^x_{i+1,i}}{\partial x_i} + \frac{\partial r_{l,m}}{\partial \Delta^x_{i,i-1}} \frac{\partial \Delta^x_{i,i-1}}{\partial x_i} + \sum_j \frac{\partial r_{l,m}}{\partial \Delta^{xx'}_{i,j}} \frac{\partial \Delta^{xx'}_{i,j}}{\partial x_i}$$

where the partial derivatives $\frac{\partial \Delta^{xx'}_{i,j}}{\partial x_i}$ and $\frac{\partial \Delta^x_{i+1,i}}{\partial x_i}$ can be directly computed depending on the cost function $\delta$.

## F. Computational Cost

As mentioned at the end of Section 5, for the (soft-)DTW and the (soft-)TWED, the computational cost for $N$ time series of size $T$ in dimension $d$, is in $\mathcal{O}(NT^2d)$ per epoch. Indeed, forward pass and backward pass for the gradient computation involves $O(T^2)$ distances computations in dimension $d$ for each sample $X$. The Figure 4 illustrates the computational time in practice for some values of $d, N, T$.

## G. Details and Additional Experiments

### G.1. Synthetic Data Generation

For synthetic experiments, we used a bipolar pulse ideal pattern. An example is presented on the bottom left of Figure 5. Regarding the warping function, we assume that the signals are discrete versions of functions $f : [0,1] \to \mathbb{R}$, we then take the points $x_1 = 0, x_2 = .25, x_3 = .5, x_4 = .75, x_5 = 1$. We then consider noisy versions of these points $y_i = x_i + u_i$ where $u_i \sim \mathcal{U}(-0.1\, w_s, 0.1\, w_s)$, with $w_s$ the warping power parameter. The $y_i$ are then projected on $[0,1]$ and sorted so that we obtain $z_1 < z_2 < z_3 < z_4 < z_5$. Then, a linear interpolation between the points $(x_i, z_i)$ is performed, resulting on a piecewise linear warped temporal line and the interpolation of the bipolar pulse is then performed on this warped temporal line. An example of warping deformations with several warping parameters is presented in Figure 5.

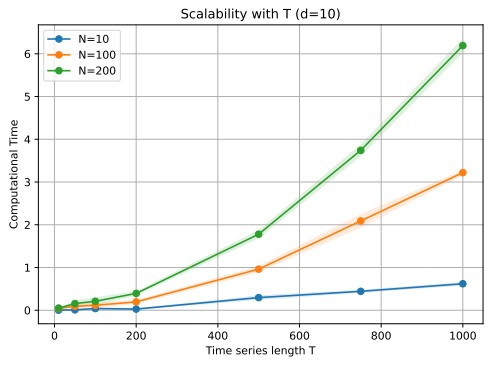 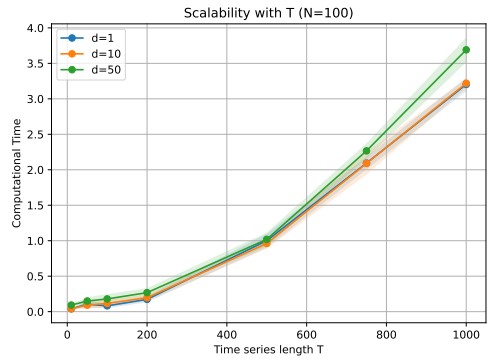

*(a)* Computational cost as a function of $N$.       *(b)* Computational cost as a function of $d$.

*Figure 4.* Computational cost of DTW with respect to the number of time series $N$, the dimension $d$ and the time series length $T$.

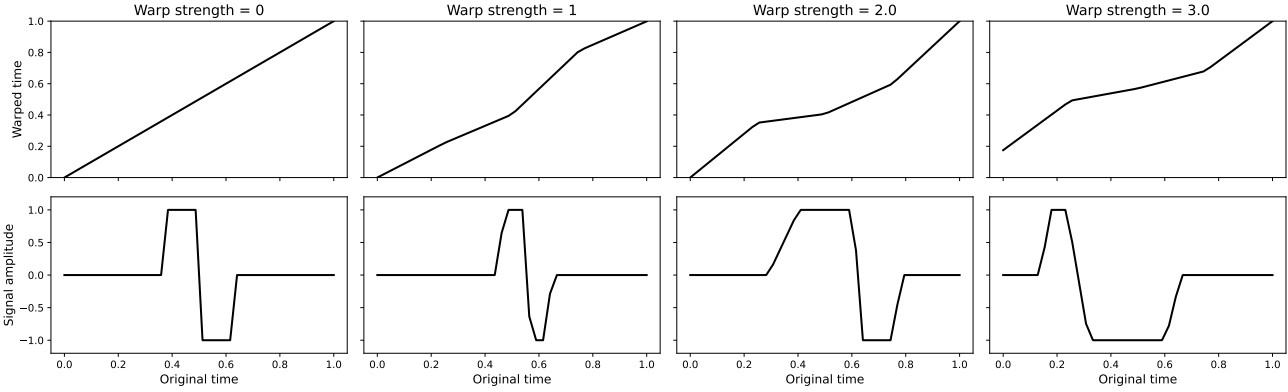

*Figure 5.* Example of synthetic data generation.

## G.2. $\theta$ Impact

In this section, we evaluate the impact of $\theta$ on the results (see Figures 6 and 7). Note that larger values of $\theta$ yield smoother soft-DTW (or soft-TWED), but also increase the discrepancy with standard DTW (or TWED). We observe that this smoothing is beneficial in settings with limited training data or highly noisy labels ($N \leq 500$ or $\tau \leq 0.8$), leading to improved accuracy and higher Kendall's tau. In contrast, smaller values of $\theta$ become more effective as the number of samples increases and when annotations are reliable.

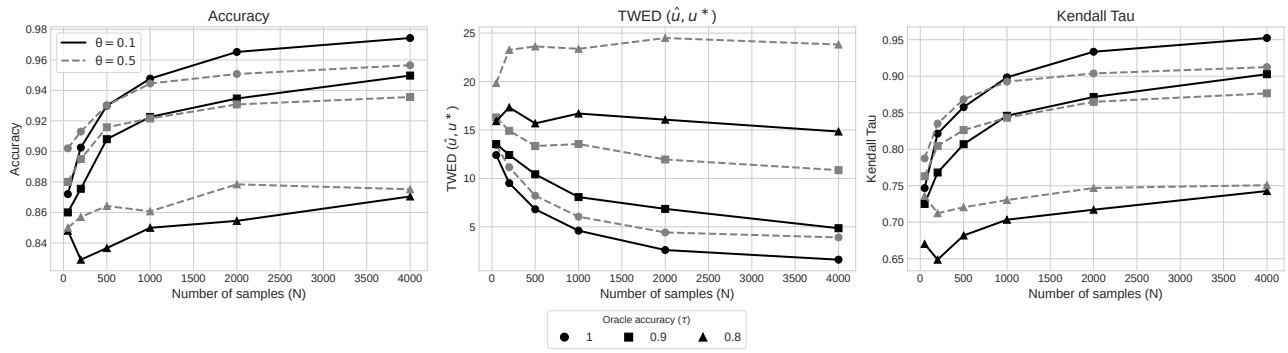

*Figure 6.* TWED $\theta$ impact.

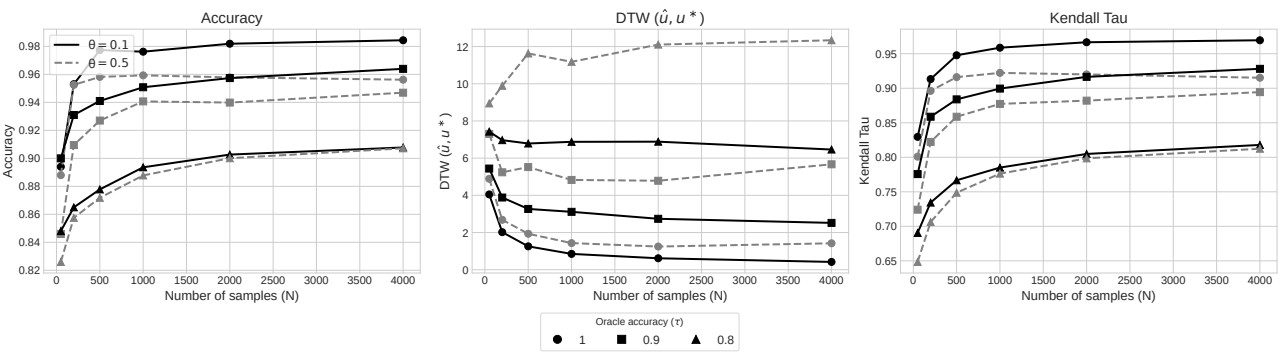

*Figure 7.* DTW $\theta$ impact.

## G.3. Full Table Synthetic Dataset

The same conclusions of Section 6.2.2 can be made looking at the Spearmann score and the Accuracy (see Table 8).

| Metric | Warp | **0** | | | **0.5** | | | **1** | | | **3** | | |
|--------|------|----|-----|------|----|-----|------|----|-----|------|----|-----|------|
| | N | **50** | **500** | **2000** | **50** | **500** | **2000** | **50** | **500** | **2000** | **50** | **500** | **2000** |
| | **Method** | | | | | | | | | | | | |
| **Kendall Tau** ↑ | DTW | 0.65 (0.13) | 0.81 (0.06) | **0.85** **(0.01)** | 0.56 (0.2) | 0.72 (0.05) | 0.76 (0.0) | 0.62 (0.05) | 0.72 (0.04) | **0.76** **(0.01)** | 0.41 (0.15) | **0.69** **(0.05)** | **0.73** **(0.02)** |
| | L2 | 0.37 (0.08) | 0.75 (0.02) | 0.83 (0.01) | 0.25 (0.04) | 0.48 (0.02) | 0.52 (0.01) | 0.24 (0.06) | 0.42 (0.01) | 0.46 (0.01) | 0.13 (0.08) | 0.23 (0.02) | 0.25 (0.02) |
| | TWED | **0.76** **(0.05)** | **0.82** **(0.01)** | 0.83 (0.0) | **0.71** **(0.04)** | **0.78** **(0.02)** | **0.8** **(0.0)** | **0.67** **(0.02)** | **0.74** **(0.01)** | 0.76 (0.02) | **0.54** **(0.09)** | 0.67 (0.03) | 0.68 (0.04) |
| **Spearmann** ↑ | DTW | 0.83 (0.14) | **0.95** **(0.04)** | **0.97** **(0.0)** | 0.74 (0.26) | 0.9 (0.05) | 0.93 (0.0) | 0.82 (0.05) | 0.91 (0.04) | **0.93** **(0.0)** | 0.57 (0.19) | **0.88** **(0.05)** | **0.91** **(0.01)** |
| | L2 | 0.53 (0.11) | 0.92 (0.01) | 0.96 (0.0) | 0.37 (0.06) | 0.68 (0.03) | 0.72 (0.01) | 0.35 (0.08) | 0.6 (0.02) | 0.65 (0.01) | 0.19 (0.11) | 0.34 (0.03) | 0.37 (0.02) |
| | TWED | **0.92** **(0.03)** | **0.95** **(0.0)** | 0.96 (0.0) | **0.89** **(0.03)** | **0.94** **(0.01)** | **0.95** **(0.0)** | **0.86** **(0.01)** | **0.92** **(0.01)** | 0.93 (0.01) | **0.73** **(0.11)** | 0.86 (0.03) | 0.87 (0.03) |
| **Accuracy** ↑ | DTW | 0.81 (0.07) | 0.9 (0.03) | **0.93** **(0.01)** | 0.79 (0.12) | 0.86 (0.04) | 0.88 (0.01) | 0.82 (0.04) | **0.86** **(0.02)** | **0.88** **(0.01)** | 0.72 (0.09) | **0.84** **(0.04)** | **0.86** **(0.01)** |
| | L2 | 0.68 (0.07) | 0.88 (0.01) | 0.91 (0.01) | 0.66 (0.05) | 0.74 (0.01) | 0.76 (0.01) | 0.62 (0.07) | 0.7 (0.02) | 0.73 (0.01) | 0.57 (0.07) | 0.6 (0.01) | 0.62 (0.02) |
| | TWED | **0.89** **(0.05)** | **0.91** **(0.01)** | 0.92 (0.01) | **0.87** **(0.04)** | **0.88** **(0.02)** | **0.9** **(0.01)** | **0.83** **(0.05)** | **0.86** **(0.02)** | **0.88** **(0.01)** | **0.77** **(0.07)** | **0.84** **(0.02)** | 0.84 (0.02) |

*Figure 8.* Additional results of Section 6.2.2. For each metric, the higher the better.

## G.4. Low Dimension Model

In this section, we consider the same generative model for the data and we are solving the optimization problem in $\mathbb{R}^r$. The idea is to understand whether a low dimension version of the ideal is able to capture the geometrical properties of the problem and retrieve the ranking. Figure 9 shows that for toy signals (bipolar pulse for instance) the low dimension model can achieve the same accuracy that the standard model. Indeed, for the DTW $r = 10$ is sufficient and for the TWED it is $r = 25$.

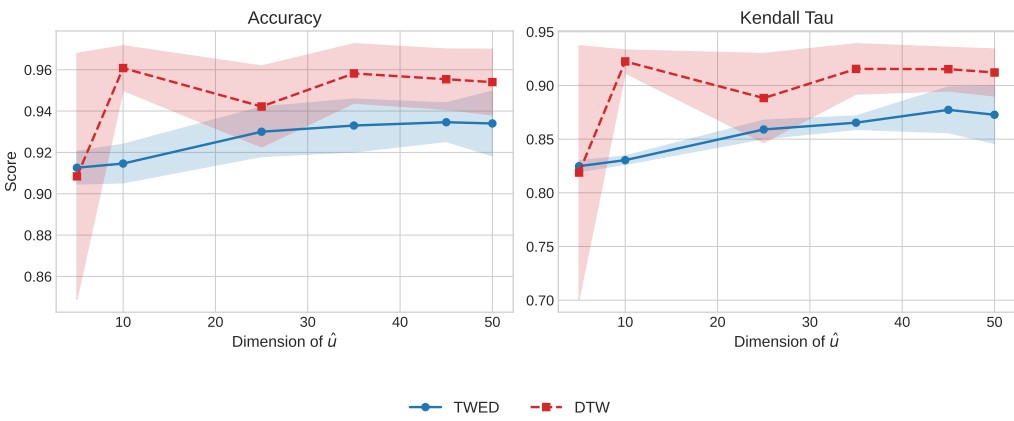

*Figure 9.* Low dimension model results.

## G.5. Motion Dataset: Additional Figures

Figure 10 illustrates the learned ideal point and the nearest and farthest signals to the ideal. We see that the estimated ideal has a waveform similar to a typical time series recording of a person walking ($y = 1$). This means that our model correctly learned a meaningful ideal.

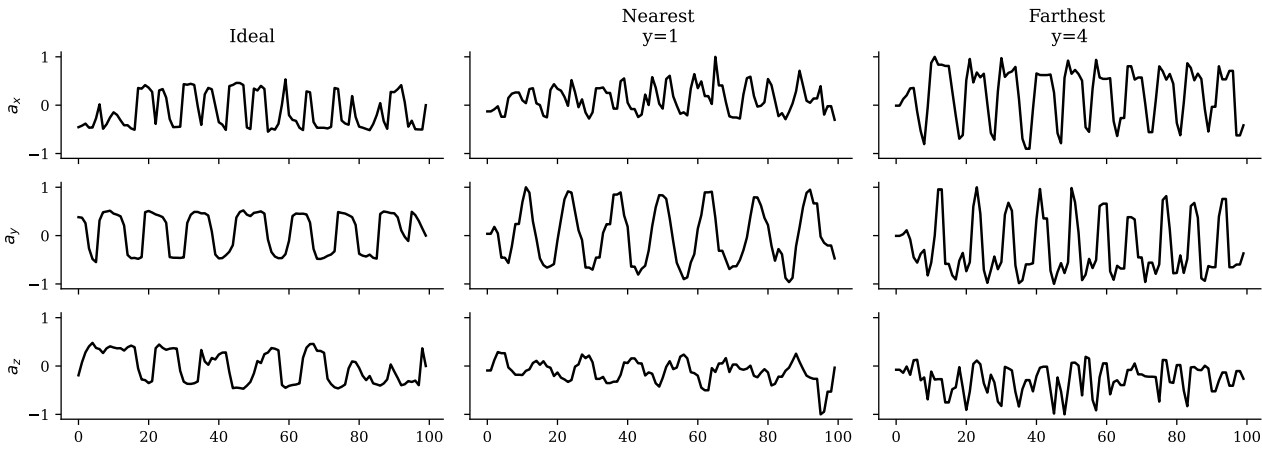

*Figure 10.* Left: estimated ideal. Middle: nearest signal to the ideal. Right: farthest signal to the ideal.

## G.6. Additional Experiments on Semi-synthetic Data

In order to strengthen the experimental section, we now add two additional experiments using semi-synthetic datasets. As common in ranking literature, these datasets are built from real data with synthetic pairwise labels. An ideal signal is randomly selected from the test set, and scores are computed via DTW distance to it.

- In the case of the handwritting dataset (Shokoohi-Yekta et al., 2017), we have 150 train cases, from which we extracted 1500 pairwise comparisons, and 850 test cases where each signal is of size 3x53 (after downsampling). The three dimensions are the three accelerometer values. To evaluate the estimated ranking we use the kendal metric (the higher the better) and obtained DTW $0.72$ / TWED $0.62$ / Rocket (2000) + RankSVM $0.59$. Furthermore, Figure 11 shows the True vs Estimated rank for our method (DTW).

- In the case of the ECG dataset (Bousseljot et al., 1995), we construct $10000$ comparisons from the $290$ signals, each of dimension $12 \times 100$ (after segmentation and downsampling). The results on this large dataset confirm that, as

expected, our method learns an ideal $\hat{u}$ that accurately ranks the time series (Figure 12 Left panel) and approximates $u^*$ (Figure 12 Right panel).

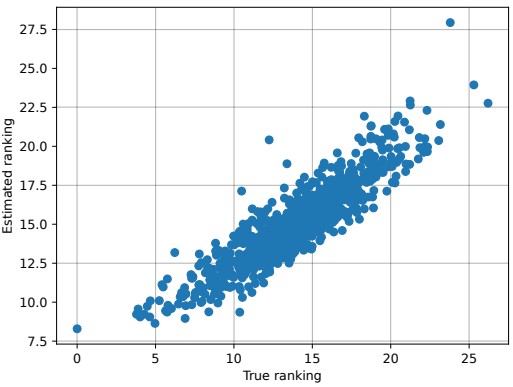

*Figure 11.* True vs Estimated rank for our method (DTW) on the handwritting dataset.

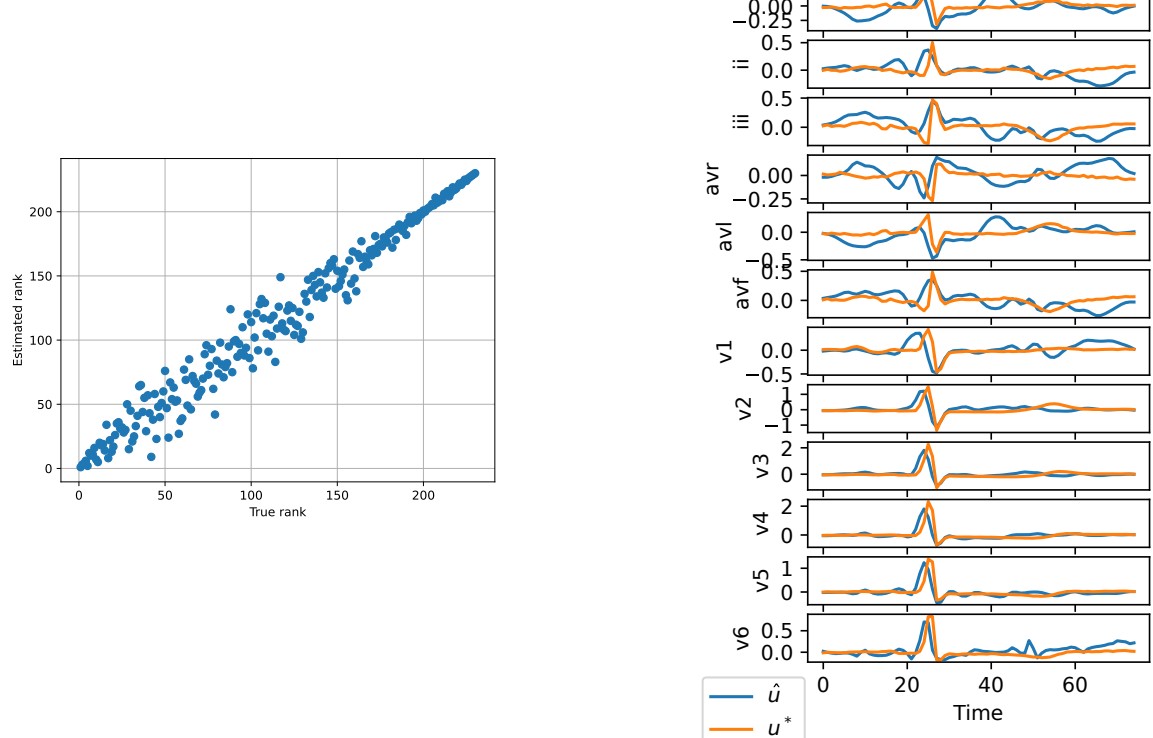

*Figure 12.* Left: True vs Estimated rank for our method (DTW) on the ECG dataset. Right: Visualisation of $\hat{u}$ and $u^*$.

## G.7. Extension to Multiple Ideal Points.

The Ideal Point Model introduced assumes that all preferences can be explained by a single ideal time series $u^*$. While this assumption leads to an interpretable and parsimonious model, it may be restrictive in applications where several distinct prototypes can explain the observed preferences. For instance, the data may contain several abnormal patterns

corresponding to different modes of variation. A natural extension is therefore to introduce $K$ ideal time series $\mathcal{U} = \{u_1, \ldots, u_K\}, u_k \in E$. Given a time series $X$, its distance to the set of ideals is defined as the distance to the closest ideal:

$$d(X, \mathcal{U}) = \min_{1 \le k \le K} d(X, u_k).$$

The pairwise preference rule then becomes

$$X \prec X' \iff d(X, \mathcal{U}) < d(X', \mathcal{U}).$$

Given pairwise observations

$$\mathcal{S} = \{(X_i, X'_i, y_i)\}_{i=1}^{n}, \qquad y_i \in \{-1, +1\},$$

we can estimate the set of ideals by empirical risk minimization:

$$\widehat{\mathcal{U}} \in \arg\min_{\mathcal{U} \subset E, \, |\mathcal{U}|=K} \frac{1}{n} \sum_{i=1}^{n} \ell\left(y_i \left[d(X_i, \mathcal{U}) - d(X'_i, \mathcal{U})\right]\right). \tag{20}$$

Equivalently, writing explicitly the minimum over the ideal points,

$$\widehat{u}_1, \ldots, \widehat{u}_K \in \arg\min_{u_1, \ldots, u_K \in E} \frac{1}{n} \sum_{i=1}^{n} \ell\left(y_i \left[\min_{1 \le k \le K} d(X_i, u_k) - \min_{1 \le k \le K} d(X'_i, u_k)\right]\right).$$

This formulation can be interpreted as a mixture-like extension of the Ideal Point Model: each signal is compared to the closest ideal, and preferences are induced by proximity to the set of prototypes. The model can therefore represent several latent notions of severity or abnormality. However, the objective above is non-smooth because of the minimum over the ideal points. To obtain a differentiable surrogate, we replace the hard minimum by a soft minimum. We define the soft distance to the set of ideals as

$$d_\tau(X, \mathcal{U}) = \mathrm{softmin}_\tau\left(d(X, u_1), \ldots, d(X, u_K)\right).$$

The differentiable empirical objective becomes

$$\widehat{\mathcal{U}}_\tau \in \arg\min_{\mathcal{U} \subset E, \, |\mathcal{U}|=K} \frac{1}{n} \sum_{i=1}^{n} \ell\left(y_i \left[d_\tau(X_i, \mathcal{U}) - d_\tau(X'_i, \mathcal{U})\right]\right).$$

When $d$ itself is replaced by a differentiable surrogate, such as soft-DTW or soft-TWED, the whole objective becomes differentiable with respect to the ideal time series $u_1, \ldots, u_K$. The parameters can then be optimized by gradient-based methods. The temperature $\tau$ controls the softness of the assignment of a time series to its closest ideal: small values of $\tau$ approximate hard nearest-ideal assignment, whereas larger values allow several ideals to contribute to the preference score. An illustrative example of first results using two ideal points is given in Figure 13. This simple example show that this extended model can learn several ideal points and thus explain preferences using several prototypes / deformations.

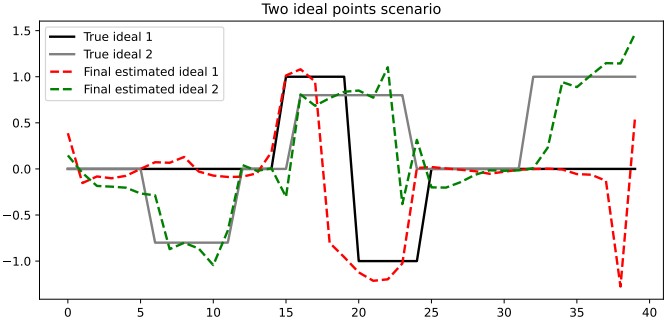

*Figure 13.* Low dimension model results.

