# OpenReview forum: "Ranking Time Series using a Time Warping Ideal Point Model"
_ICML.cc/2026/Conference — ICML 2026 spotlight_

### Official Review · Reviewer_irG3 · 2026-03-12

**Soundness:** 4
**Presentation:** 4
**Significance:** 3
**Originality:** 3
**Overall Recommendation:** 5
**Confidence:** 3

**Summary:**

In this paper, the authors propose an Ideal Point Model (IPM) for ranking time series from pairwise comparisons, leveraging elastic distances (DTW and TWED) to achieve robustness to temporal misalignments while maintaining interpretability. They validate their approach using synthetic and real data and report that it produces accurate, robust rankings under noisy annotation conditions.

**Compliance With Llm Reviewing Policy:**

Affirmed.

**Final Justification:**

The authors addressed all of the minor issues raised. They supported the rebuttal with proper justifications, new results, and illustrations, thereby improving an already solid paper. The most significant addition was the inclusion of two experiments using semi-synthetic datasets, which addressed the most critical issue of limited real-data evaluation. Thus, I increased the rating in "Soundness."

**Key Questions For Authors:**

None

**Limitations:**

The authors acknowledge some limitations in the conclusion, but the paper could benefit from addressing the aforementioned minor issues as limitations as well.

**Strengths And Weaknesses:**

Soundness.
The paper is technically sound. The method is theoretically grounded. The methodology is clearly formalized with well-defined equations. The evaluation is also comprehensive.
- Minor issues:
1. The authors claim that pairwise annotations generally result in higher-quality supervision. However, this statement is questionable for certain tasks. For instance, consider the detection of abnormal events in physiological signals, as discussed in the paper. A good example is the detection of epileptic seizures in EEGs. Experts often disagree about "transitional states" between normal activity and seizures, which leads to inconsistent seizure onset detection. It is difficult to determine whether pairwise annotations would resolve this particular issue.
2. The model assumes one global ideal time series that explains all preferences, and the authors acknowledge this as a limitation. In a real-world scenario, different annotators may have different preferences. Returning to the seizure detection example, different clinicians may annotate different parts of the EEG signal as seizure onset. The ground truth here can be highly debatable.
3. The method lacks any kind of ranking for annotators. In real life, some annotators may be more proficient than others, so their annotations should probably carry more weight.
4. The authors acknowledge the computational cost associated with TWED and DTW for long time series but do not report training times or compare the computational cost to baselines.
5. Synthetic data is generated by warping and adding noise to the ideal time series used for evaluation. This generation approach may favor the method. Additionally, real-life data distortions can be far more complex.
6. Real dataset evaluation is limited. The dataset is small, and the ranking task is artificially constructed by permuting class labels.

Presentation.
The submission is clearly written and well-structured. Introduction properly positions the work in the context of prior studies. The paper provides sufficient detail to facilitate reproducibility.

Significance.
This paper addresses the important issue of expert annotations in time series. The scope of its impact is quite broad because agreement between expert annotations is important for a wide variety of datasets.

Originality.
This work introduces a new, theoretically grounded method that has been evaluated on both synthetic and real data.

---

> ### Author Rebuttal · Authors · 2026-03-30
>
> We thanks the reviewer for the comments on our paper.
>
> **1) Pairwise annotations lead to higher-quality supervision:**
>
> We agree that pairwise comparisons are not suitable for all tasks, including some segmentation problems. However, they have shown strong performance across modalities (image, text, sound) [1-3]. In physiological time series, we observed high agreement on clearly normal or abnormal signals, but significant disagreement on borderline cases (inter-annotator F1 < 0.5 in some settings). Pairwise comparisons can mitigate this issue, as annotators more reliably assess relative abnormality than absolute labels. Similar improvements in agreement have been reported in medical imaging [4].
>
> **2) Single ideal point:**
>
> We acknowledge that assuming a single ideal point is a limitation as in practice, annotators may rely on multiple reference patterns. We are currently extending the model to multiple ideals by incorporating distances to several points $u^\ast_1, \ldots, u^\ast_k$, controlled in the loss via a softmin weighting. Preliminary results with two ideals are promising (see https://ibb.co/hxdbM2RM), although identifiability theorems remains a challenge. Regarding inter-clinician variability, the model can be applied per annotator to recover individual ideal points, providing insights into their criteria and disagreements.
>
> **3) Annotators may be more proficient than others:**
>
> We agree that annotator expertise should be accounted for. The model can be trained per annotator or jointly. In the latter case, a weighted loss could incorporate annotator reliability if such information is available, as explored in [5] for instance. This is an interesting direction for future work.
>
> **4) Computational cost:**
>
> For $N$ time series of size $T$ in dimension $d$, the cost is in $O(N T^2 d)$ per epoch. An illustration is available here: https://ibb.co/dNCx8sf. Note that it exists other versions of the DTW that explore only a small part of the possible temporal alignments, leading to a lower computational cost [6]. We will also add a discussion on this in the final version.
>
> **5) Synthetic data may favor the method / real-life distortions:**
>
> We agree that the synthetic setting favors our model, as it follows the IPM assumptions. Our goal here is not to claim superiority, but to validate the optimization approach (gradient descent with soft-DTW/TWED).
> Regarding real distortions, DTW handles time shifts and dilations but not arbitrary transformations. We emphasize that the choice of metric should reflect the task: invariance should be limited to relevant transformations. DTW and TWED are standard and, in general, appropriate choices for time series.
>
> **6) Real dataset evaluation**
>
> As already said, while pairwise comparisons are widely used to improve annotation robustness across domains [1-3], this framework has not yet been explored for time series. This both motivates our approach and explains the lack of existing datasets with native pairwise annotations. The main point of this paper is the introduction of the IPM with elastic distances and its theoretical foundations. We hope that our results, together with successes in other domains, will encourage the development of such datasets.
>
> In order to strengthen the experimental section, we will add two additional experiments using semi-synthetic datasets. As common in ranking literature, these datasets are built from real data with synthetic pairwise labels. An 'ideal signal' is randomly selected from the test set, and scores $y_i$ are computed via DTW distance to it.
>
> - Handwritting data [7]: we have 150 train cases, from which we extracted 1500 pairwise comparisons, and 850 test cases where each signal is of size 3x53 (after downsampling). As in the paper, to evaluate the estimated ranking we use the kendall metric (the higher the better) and obtained DTW 0.72 / TWED 0.62 / Rocket (2000) + RankSVM 0.59. Furthermore, https://ibb.co/V0z9rdgw shows the True vs Estimated rank for our method (DTW).
>
> - ECG data [8]: we make 10000 comparisons of the 290 signals, each of dimension 12x100 (after segmentation and downsampling). Results on this large dataset confirm that our method learns an ideal $\hat{u}$ that accurately ranks the time series (see Figure https://ibb.co/KzNJhRVB) and approximates $u^∗$ (see Figure https://ibb.co/8nkGNCPL).
>
> ---
> [1] Lee et al. Image aesthetic assessment based on pairwise comparison
>
> [2] Guo, Tianyu, et al. "Brace: A benchmark for robust audio caption quality evaluation
>
> [3] Stiennon, Nisan, et al. Learning to summarize with human feedback
>
> [4] Jang, I. et al. Decreasing annotation burden of pairwise comparisons with human-in-the-loop sorting
>
> [5] Saad, El Mehdi et al. Active ranking of experts based on their performances in many tasks
>
> [6] Hiroaki Sakoe et al. Dynamic programming algorithm optimization for spoken word recognition
>
> [7] M. Shokoohi-Yekta, B. Hu, J. Wang and E Keogh (2017)
>
> [8] www.physionet.org/content/ptbdb/1.0.0/

---

> > ### Author Rebuttal · Reviewer_irG3 · 2026-04-02
> >
> > Thank you for your exhaustive response. I consider all issues resolved.

---

> > > ### Author Response · Authors · 2026-04-07
> > >
> > > Thank you for your constructive review. We are glad the rebuttal helped address your concerns.

---

### Official Review · Reviewer_wo17 · 2026-03-12

**Soundness:** 3
**Presentation:** 3
**Significance:** 2
**Originality:** 3
**Overall Recommendation:** 4
**Confidence:** 3

**Summary:**

This paper explores time series supervision in the context of a ranking learning problem leveraging comparative pairwise annotations, framed as a more robust alternative to traditional classification approaches and labels (especially under noisy label settings). They leverage an Ideal Point Model (IPM), but unlike previous works which have largely focused on embedding items in a latent space, this work treats the ideal point as a singular learned time series. They utilize elastic-time distances/similarity measures, specifically introducing soft-TWED, a differentiable analogy of the Time Warp Edit Distance (TWED) which replaces TWED’s min with a softmin, to enable gradient descent. They contrast this with soft-DTW, a differentiable elastic similarity measure from past work, and show that their method with soft-DTW or soft-TWED achieves higher performance than L2-distance IPM model, as well as one based on the RankNet/Catch22 baseline.

**Compliance With Llm Reviewing Policy:**

Affirmed.

**Final Justification:**

The authors have addressed all my comments which significantly strengthened the paper and thus I'm raising the score from Weak Reject to Weak Accept. I still have reservations about novelty thus I'm scoring it 4 rather than 5. This is my final recommendation.

**Key Questions For Authors:**

Are there no larger real-world time series datasets with natively available ranking labels that you could have used? If there aren’t, does this harm the relevance of your work?

Would you consider better motivating the practical utility of your method by comparing it against supervised classification models? Why or why not?

You mention the limitation of larger computational cost for longer time series, but what is the computational expense given larger datasets?

**Limitations:**

Since the authors' model operates directly in the waveform space and the “ideal” is itself an explicit time series, the robustness of their method extends only to the specific temporal distortions that elastic similarity measures can absorb. While the paper emphasizes robustness to time-warping deformations such as shifts and dilations, this can still be brittle on real signals whose superficial variation is not well handled by endpoint-anchored elastic alignment—for example, two ECG snippets that are clinically equivalent may appear dissimilar to this method if they are arbitrarily windowed at different phases of the cardiac cycle, which is precisely why beat segmentation/alignment is often treated as essential preprocessing and why cyclic variants of DTW have been proposed. In that sense, they may be understating the rigidity of their approach: Because the model ranks against a literal waveform template rather than a more abstract latent representation, superficial differences may still be mistaken for meaningful deviation, limiting clean applicability to carefully preprocessed settings (such as beat-aligned template matching with ECGs). I would be curious to hear which real-world settings they believe they would perform well in, with or without domain-specific preprocessing and whether they generally have a discussion/rebuttal around learning in the input space rather than latent space.

**Strengths And Weaknesses:**

Soundness
The work appears theoretically sound, with a rigorous mathematical formulation including definitions, theorems, propositions, lemmas, and corollaries that are well organized and appear correct (as per my limited familiarity with IPMs and elastic similarity measures).
However, they seem to focus on demonstrating that their method is functional over synthetic data rather than empirically valuable through real-data validation. Namely, they leverage a single, exceedingly small real-world time series BasicMotions dataset (training on 40 signals and testing on 40 signals) without native comparative pairwise annotations, where they must induce an order on the labels. While deriving pairwise supervision from existing signals is common in ranking, doing so here from nominal activity classes rather than finding a dataset supporting a more naturally ordered real-world ranking problem is confusing, and may limit how strongly the results support the method’s practical utility.
Beyond the real-world dataset, both their feature-based ranking baseline model, RankNet, and their time-series feature extractor, Catch22, did not seem especially strong. It appears they could have used newer or stronger alternatives, such as RankSVM and LambdaMART-style tree rankers, and ROCKET/MiniRocket for their feature extractor, where ROCKET actually reported Catch22 as the least accurate method among the compared time-series baselines in that benchmark. In brief, their choice of a simple baseline makes me question whether their method is particularly relevant outside of their particular ranking niche.

Presentation
The authors strive to outline an important topic and analyze a central concept with clear high-level structure. The paper is easy to follow and proceeds logically. Their mathematical formulation is quite readable. That said, the paper’s motivation and practical framing are somewhat underspecified: the introduction motivates pairwise comparisons as a more reliable alternative to standard supervised labeling, but the empirical section does not directly test that broader claim, which makes the overall narrative feel looser than it could be. The figures are informative but not self-contained, relying heavily on the surrounding text to explain what is being shown. I would encourage the authors to expand the captions and strengthen the clarity of their motivation/novelty.

Significance
Since the authors' motivation is at times unclear, the problem setting seems rather niche as a result. For example, they question the reliability of supervised classification models, but do not venture beyond loose motivational framing into empirical proof. It may then be difficult for an audience unfamiliar with ranking-based learning to care about the setting in which they operate, especially considering that the arguably limited applicability of their method under the rare availability of comparative pairwise annotations in real-world datasets and lack of discussion around inducing ranking labels from more traditionally available time series labels.
Part of their contribution appears applying soft-DTW to the IPM setting, as this does not appear to have already been done. Soft-DTW and soft-TWED aren’t presented as competing since they are differentiable surrogates for different underlying distances. And from what I can gather, their TWED-based model has theoretical properties that the DTW-based model does not fully enjoy (e.g., Remark 4.2). However, I don’t see any discussion showing that soft-TWED is advantageous (universally or conditionally) over soft-DTW, which could beg the question as to whether we needed this new differentiable elastic distance metric? Their metrics do seem better, at times, but I believe some discussion on this may be valuable to establish the significance of introducing soft-TWED.

Originality
The paper has some originality to it. The authors introduce soft-TWED a differentiable analogy of TWED and apply both it and soft-DTW to the IPM setting in such a way that does not appear to have already been done. The fact that they treat the ideal point itself as an explicit time series is certainly original, though perhaps not well-motivated as an input space approach compared to latent space approaches given its potential rigidity.

---

> ### Author Rebuttal · Authors · 2026-03-30
>
> We thank the reviewer for their feedback and interesting questions.
>
> **1) Real data and native annotation:**
>
> Pairwise comparisons are widely used to improve annotation robustness across domains, including images [1], audio evaluation [2], and LLM-generated text [3]. To our knowledge, this framework has not yet been explored for time series. This both motivates our approach and explains the lack of existing datasets with native pairwise annotations. The main point of this paper is the introduction of the IPM with elastic distances and its theoretical foundations. We hope that our preliminary results, together with successes in other domains, will encourage the development of such datasets, which we believe are highly valuable for the medical time series community.
>
> In order to strengthen the experimental section, we will add two additional experiments using semi-synthetic datasets. As common in ranking literature, these datasets are built from real data with synthetic pairwise labels. An 'ideal signal' is randomly selected from the test set, and scores $y_i$ are computed via DTW distance to it.
>
> - Handwritting data [4]: we have 150 train cases, from which we extracted 1500 pairwise comparisons, and 850 test cases where each signal is of size 3x53 (after downsampling). The three dimensions are the three accelerometer values. As in the paper, to evaluate the estimated ranking we use the kendal metric (the higher the better) and obtained DTW 0.72 / TWED 0.62 / Rocket (2000) + RankSVM 0.59. Furthermore, https://ibb.co/V0z9rdgw shows the True vs Estimated rank for our method (DTW).
>
> - ECG data [5]: we make 10000 comparisons of the 290 signals, each of dimension 12x100 (after segmentation and downsampling). The results on this large dataset confirm that, as expected, our method learns an ideal $\hat{u}$ that accurately ranks the time series (Fig. https://ibb.co/KzNJhRVB) and approximates $u^∗$ (Fig. https://ibb.co/8nkGNCPL).
>
> **2) RankSVM, LambdaMART, and ROCKET:**
>
> Recall that LambdaMART optimizes ranking from relevance scores rather than pure pairwise comparisons. It therefore seems to us that this is not necessarily an appropriate baseline contrary to rankSVM, which we added. We also compared a CNN-based RankNet and ROCKET-based pipelines. Results are in the Table here: https://ibb.co/GQmjgdzb. They are consistent with those reported in the paper, except for ROCKET(1000), which improves performance but reduces interpretability due to its random features.
>
> **3) Clarity of images:**
>
> We will improve figures readability as suggested.
>
> **4) Reliability of classifier:**
>
> This is an interesting remark. There are several explanations for the unreliability of classifiers. In medical time series, binary labels often show low inter-annotator agreement, whereas pairwise comparisons are more consistent [6]. Moreover, pairwise comparisons could lead to better models than ones trained on binary annotations. To justify this point, we will add an illustrating experiment. Taking the semi-synthetic ranking dataset built using handwriting dataset as explained in point 1), we compared two algorithms:
> - We assign label 0 to the bottom 50% ranked samples and 1 to the rest, extract features with ROCKET(5000), and train an XGBoost classifier, achieving 74% test accuracy.
> - Using 1500 pairwise comparisons from the same training set, we extract ROCKET(5000) features and train RankSVM. A threshold selected on the training set is then used for classification, yielding 81% test accuracy.
>
> Thus, here, learning to rank leads to a better post-classification (see https://ibb.co/zVhBZj34).
>
> **5) TWED vs DTW:**
>
> TWED is a proper metric, enabling stronger theoretical guarantees (e.g., identifiability of $u^\ast$) and leading to a smoother loss function (better Lipschitz constants). On the other hand, DTW, while not a proper metric, is widely used and efficient in practice. Note that, most of our theoretical results hold for both (except identifiability), and empirically their performance are similar for the datasets we have tested. Finally, note also that if $u^* \in \mathbb{R}^{T \times d}$ verifies that for all $t \in [1,T-1], u[t] \neq u[t+1]$, then Assumption 4.9 can also be satisfied with the DTW, implying the identifiability.
>
> **6) Scalability**
>
> For $N$ time series of size $T$ in dimension $d$, the cost is in $O(N T^2 d)$ per epoch. An illustration is available here: https://ibb.co/dNCx8sf.
>
> ---
> [1] Lee et al. "Image aesthetic assessment based on pairwise comparison a unified approach to score regression, binary classification, and personalization."
>
> [2] Guo, Tianyu, et al. "Brace: A benchmark for robust audio caption quality evaluation."
>
> [3] Stiennon, Nisan, et al. "Learning to summarize with human feedback."
>
> [4] M. Shokoohi-Yekta, B. Hu, J. Wang and E Keogh (2017)
>
> [5] https://www.physionet.org/content/ptbdb/1.0.0/
>
> [6] Jang, I. et al. Decreasing annotation burden of pairwise comparisons with human-in-the-loop sorting.

---

> > ### Author Rebuttal · Reviewer_wo17 · 2026-04-03
> >
> > This is a pretty strong rebuttal and the additions add a lot. I will change my score from Weak Reject to Weak Accept as the novelty is not significantly improved (that's a property of the idea itself) thus it's only Weak Accept rather than Accept. Thank you for doing all this work and good luck!

---

> > > ### Author Response · Authors · 2026-04-07
> > >
> > > Thank you for your constructive review. We are glad the rebuttal helped address your concerns.

---

### Official Review · Reviewer_q33M · 2026-03-13

**Soundness:** 2
**Presentation:** 4
**Significance:** 3
**Originality:** 3
**Overall Recommendation:** 5
**Confidence:** 4

**Summary:**

This paper addresses the challenge of low inter-annotator agreement in expert-labeled time series classification by reformulating the task as a pairwise ranking problem. The authors propose an Ideal Point Model (IPM) where time series are ranked based on their distance to a latent "ideal" reference signal. To account for temporal distortions such as shifts and dilations, the framework replaces standard Euclidean distances with elastic metrics, specifically Dynamic Time Warping (DTW) and Time Warp Edit Distance (TWED). The authors provide theoretical guarantees, including Lipschitz continuity and convergence bounds, and introduce a differentiable "soft-TWED" to enable end-to-end gradient-based optimization.

**Compliance With Llm Reviewing Policy:**

Affirmed.

**Final Justification:**

The authors presented a strong rebuttal that has successfully addressed all of my concerns, particularly increasing the soundness and clarity of the paper. Therefore, I have raised my score from 4 to 5.

**Key Questions For Authors:**

1. How do you reconcile the use of DTW with Assumption 4.9, given that DTW's lack of metric properties, specifically the identity of indiscernibles, technically violates the uniqueness of the ideal point $u^\ast$?
2. Can you justify the choice of baselines in the real-world experiments? Specifically, why compare against a feature-extracted RankNet instead of an end-to-end deep time series embedding model trained directly on the raw signals with a pairwise ranking loss?
3. How does the model behave when applied to datasets with highly multi-modal "ideal" states? Have you experimented with extending the framework to a mixture of ideal points to prevent the model from learning an unrepresentative average signal?

**Limitations:**

yes

**Strengths And Weaknesses:**

- Strengths:
  - Relevance and Motivation: Shifting from absolute classification to relative pairwise comparisons is a highly practical and well-motivated strategy for domains with subjective evaluation criteria, such as medicine.
  - Theoretical Rigor: Bridging IPMs with elastic metrics is a mathematically sound contribution. The proofs for Lipschitz continuity and excess risk bounds provide a strong theoretical foundation for the proposed models.
  - Methodological Innovation: The introduction of the differentiable soft-TWED is a valuable technical contribution that facilitates gradient descent optimization for a true metric.

- Weaknesses:
  - The Single Ideal Assumption: The model rigidly assumes a single ideal time series $u^\ast$. In complex real-world datasets, "ideal" states are frequently multi-modal [1, 2]. Forcing the model to learn a singular point risks converging on an averaged, nonsensical signal that does not represent any valid physiological or mechanical state.
  - Identifiability Contradictions with DTW: The theoretical guarantees rely heavily on Assumption 4.9 (positive measure of distinguishing pairs) to ensure the unique recovery of $u^\ast$. However, the authors concede that DTW is not a proper metric; any time-warped version of the ideal point yields a distance of zero. This lack of identifiability directly violates the uniqueness assumption, which could lead to unstable optimization landscapes.
  - Computational Scalability: Both DTW and TWED require dynamic programming with an $O(N^2)$ computational complexity. While the authors introduce a "low dimension model" to compress the ideal signal, it acts as a workaround rather than a fundamental solution to the heavy computational burden for large datasets with long sequences.
  - Weak Empirical Baselines: In the Basic Motions experiment, the proposed method is compared against RankNet applied to Catch22 extracted features rather than raw time series. Comparing a raw-signal elastic distance model against a feature-extracted baseline is an inherently unfair evaluation. Modern deep-learning baselines operating directly on time series should be included.

- Soundness: The paper is technically sound, with rigorous theoretical analysis and well-designed experiments. However, the reliance on a single ideal point and the use of DTW raise concerns about the practical identifiability and stability of the model.
- Presentation: The paper is well-written and structured, with a clear narrative and proper contextualization within the literature. The theoretical contributions are clearly articulated, and the experimental results are presented in a comprehensible manner.
- Significance: The paper promotes a novel perspective on time series classification that could have significant implications for domains where expert labeling is subjective. However, the practical impact may be limited by the assumptions and computational challenges identified.
- Originality: The methods and ideas presented are original, particularly the integration of elastic metrics into the IPM framework and the development of the soft-TWED.

1. Multidistribution Time-Series Prototype Learning for Crop Mapping With Sentinel-1 SAR Imagery, IEEE Transactions on Geoscience and Remote Sensing
2. Repurposing Foundation Model for Generalizable Medical Time Series Classification, ICLR 2026

---

> ### Author Rebuttal · Authors · 2026-03-30
>
> We thank the reviewer for their feedback and questions. We now address each point in detail.
>
> **1) Single ideal point and behavior with multi-modal ideal:**
>
> This is a very interesting question. It is true that if there are several ideal points, the IPM model (independently if it is applied in the standard case $R^d$ or for time series) is not adapted. Extending our framework to the "multiple ideal points" model is an exciting future research lead for us. Although it would be unrealistic to resolve this issue during one rebuttal week, we would nevertheless like to share with you our thoughts and the preliminary findings we have on this matter.
>
> One possibility to extend our model to multiple ideal points $u^\ast_1, \ldots, u^\ast_k$ could be to incorporate a distance in the loss function of Eq. (3) that controls the distance of $x_i$ to all the ideal points. We have conducted a preliminary study using a weighted sum of the distances to each ideal (using the softmin function for relevant weighting and differentiability). A primilarly result when there are two ideals is given here: https://ibb.co/hxdbM2RM. It should be noted that, following this study, we suspect that one of the difficulties will lie in identifying the $k$ ideal points because of a potential unidentfiability. Of course, it would take a lot more work to explore this issue in greater depth.
>
> **2) DTW, TWED, and Assumption 4.9:**
>
> We first would like to recall that all the theoretical results of Section 4 are valid for both DTW and TWED. This means that whatever the metrics we use, we end up with a minimizer of $R(\cdot)$ when the number of comparisons goes to infinity.
>
> Concerning the identifiability of $u^\ast$, note that Assumption 4.9 is related both to the metric (DTW or TWED) and the distribution of the data. Thus, if the dataset “variability” is poor, $u^\ast$ won’t be identifiable even with the TWED metric. On the other hand, if the data variability is sufficient, the following simple assumption on the ideal point is sufficient to prove the identifiability with DTW: let $u \in \mathbb{R}^{T \times d}$ be the ideal point. If for all $t \in [1,T-1], u[t] \neq u[t+1]$, then the assumption 4.9 can be satisfied with the DTW (actually, we enforce the ideal point to be at a positive distance of all points in $R^{T \times d}$ for the DTW). We will add a discussion on this in the final version.
>
> **3) Computational scalability:**
>
> For $N$ time series of size $T$ in dimension $d$, the cost is in $O(N T^2 d)$ per epoch. An illustration is available here: https://ibb.co/dNCx8sf. Note that it exists other versions of the DTW that explore only a small part of the possible temporal alignments, leading to a lower computational cost [1, 2].  We will also add a discussion on this in the final version.
>
> **4) Weak empirical baselines:**
>
> Based on your comment, we have added a CNN version of RankNet to work directly on time series and rankSVM. Also, we used the Rocket feature extractor (which is SOTA for time series) coupled with RankNet. Results are in the table above and are in accordance with the ones in the paper, except for the latter which is better, at the cost of interpretability since the extracted features are random.
>
> |  | Catch22 L2 | Catch22 RankNet | Catch22 RankSVM | RankNet CNN | RankNet Rocket (200) | RankSVM Rocket (200) | RankNet Rocket (1000) | RankSVM Rocket (1000) |
> | --- | --- | --- | --- | --- | --- | --- | --- | --- |
> | Kendall | 0.265 ± (0.299) | 0.608 ± (0.101) | 0.655 ± (0.096) | 0.659 ± (0.123) | 0.583 ± (0.076) | 0.646 ± (0.073) | 0.664 ± (0.097) | 0.836 ± (0.020) |
>
> **5) Why this choice of baselines?**
>
> As said above, in the experiments, we effectively have made the choice to use the most standard ranking algorithm that exists which is RankNet. The idea was to show that the ideal point model returns competitive results but also output an ideal point which can be used for interpretability (a valuable information in medicine for instance that is not provided by rankNet or rankSVM). Again for interpretability, we used the catch22 features extractions which provide explainable features. This contrasts with Rocket which generates random features that are impossible to analyze.
>
> ---
> [1] Fumitada Itakura. Minimum prediction residual principle applied to speech recognition. IEEE Transactions on Acoustics, Speech and Signal Processing, 1975.
>
> [2] Hiroaki Sakoe & Seibi Chiba. Dynamic programming algorithm optimization for spoken word recognition. IEEE Transactions on Acoustics, Speech and Signal Processing, 1978

---

> > ### Author Rebuttal · Reviewer_q33M · 2026-03-31
> >
> > Thank the author for their reply. It has addressed all of my concerns, thus I have raised my score.

---

> > > ### Author Response · Authors · 2026-04-07
> > >
> > > Thank you for your constructive review. We are glad the rebuttal helped address your concerns.

---

### Official Review · Reviewer_Znsy · 2026-03-13

**Soundness:** 2
**Presentation:** 2
**Significance:** 2
**Originality:** 2
**Overall Recommendation:** 4
**Confidence:** 2

**Summary:**

The paper proposes an ideal point model for ranking time series from pairwise comparisons using elastic distances (DTW, TWED). It proves Lipschitz continuity of both distances, derives generalization and recovery guarantees, introduces a differentiable soft-TWED, and validates the approach on synthetic and a real dataset.

**Compliance With Llm Reviewing Policy:**

Affirmed.

**Final Justification:**

The authors have addressed my concerns, and I have raised my confidence to 2.

**Key Questions For Authors:**

Refer to the weaknesses

**Limitations:**

Yes

**Strengths And Weaknesses:**

[Strengths]
- Well-motivated combination of pairwise ranking and elastic time series distances, with a specific medical annotation use case.
- Strong theoretical contributions with tight Lipschitz bounds for DTW/TWED, generalization bounds, and almost sure convergence of the estimated ideal point.
- Methodological contribution with the soft-TWED with backward gradient recursion.
- Synthetic experiments demonstrate theoretical predictions and demonstrate robustness over L2 under time warping.


[Weaknesses]
- Evaluation is limited to a synthetic dataset and a single real-world dataset, which may question generalizability.
- There is a lack of demonstration on actual scenarios, such as medical time series with real pairwise annotations needed, as highlighted in the motivation
- Comparison with only a few standard baselines and lacking computational cost and scalability analysis.

---

> ### Author Rebuttal · Authors · 2026-03-30
>
> We sincerely thank the reviewer for the constructive feedback. Below, we address each point in detail.
>
> **1) Limitation of the evaluation and medical time series with pairwise annotations**
>
> Pairwise comparisons are widely used to improve annotation robustness across domains, including images [1], audio evaluation [2], and LLM-generated text [3]. To our knowledge, this framework has not yet been explored for time series. This both motivates our approach and explains the lack of existing datasets with native pairwise annotations.
> The main point of this paper is the introduction of the IPM with elastic distances and its theoretical foundations. We hope that our preliminary results, together with successes in other domains, will encourage the development of such datasets, which we believe are highly valuable for the medical time series community.
>
> Nevertheless, as suggested, in order to strengthen the experimental section, we will add two additional experiments using semi-synthetic datasets. As it is generally the case in the literature on ranking, these datasets are constructed from real-world data, but with synthetic pairwise comparisons. More specifically, we use handwriting and ECG datasets from [4, 5]. To construct the final dataset, we randomly select an ‘ideal signal’ from the original dataset (in the test set) and compute the labels $y_i$ based on the DTW distance from this ideal:
>
> - In the case of the handwritting dataset [4], we have 150 train cases, from which we extracted 1500 pairwise comparisons, and 850 test cases where each signal is of size 3x53 (after downsampling). The three dimensions are the three accelerometer values. As in the paper, to evaluate the estimated ranking we use the kendal metric (the higher the better) and obtained DTW 0.72 / TWED 0.62 / Rocket (2000) + RankSVM 0.59. Furthermore, a figure illustrating the True vs Estimated rank for our method (DTW) is given in https://ibb.co/V0z9rdgw.
>
> - In the case of the ECG dataset [5], we make 10000 comparisons of the 290 signals, each of dimension 12x100 (after segmentation and downsampling). The results on this large dataset confirm that, as expected, our method successfully learns a function $\hat{u}$ that accurately ranks the time series (see Figure https://ibb.co/KzNJhRVB) and approximates $u^∗$ (see Figure https://ibb.co/8nkGNCPL).
>
> **2) Comparison with few baselines**
>
> Based on your remark, we have added a CNN version of RankNet to work directly on time series and rankSVM. We also used the Rocket feature extractor (which is SOTA for time series) coupled with RankNet/SVM. Results are in the table below and are in accordance with the ones in the paper, except for the latter which is better, at the cost of interpretability since the extracted features are random.
>
>
> |  | Catch22 L2 | Catch22 RankNet | Catch22 RankSVM | RankNet CNN | RankNet Rocket (200) | RankSVM Rocket (200) | RankNet Rocket (1000) | RankSVM Rocket (1000) |
> | --- | --- | --- | --- | --- | --- | --- | --- | --- |
> | Kendall | 0.265 ± (0.299) | 0.608 ± (0.101) | 0.655 ± (0.096) | 0.659 ± (0.123) | 0.583 ± (0.076) | 0.646 ± (0.073) | 0.664 ± (0.097) | 0.836 ± (0.020) |
>
> **3) Lacking computational/scalability cost**
>
> For $N$ time series of size $T$ in dimension $d$, the cost is in $O(N T^2 d)$ per epoch. An illustration is available here: https://ibb.co/dNCx8sf. Note that it exists other versions of the DTW that explore only a small part of the possible temporal alignments, leading to a lower computational cost [6, 7]. We will add a discussion on this in the final version.
>
> ---
> [1] Lee, Jun-Tae, and Chang-Su Kim. "Image aesthetic assessment based on pairwise comparison a unified approach to score regression, binary classification, and personalization."
>
> [2] Guo, Tianyu, et al. "Brace: A benchmark for robust audio caption quality evaluation."
>
> [3] Stiennon, Nisan, et al. "Learning to summarize with human feedback."
>
> [4] M. Shokoohi-Yekta, B. Hu, J. Wang and E Keogh (2017)
>
> [5] https://www.physionet.org/content/ptbdb/1.0.0/
>
> [6] Fumitada Itakura. Minimum prediction residual principle applied to speech recognition. IEEE Transactions on Acoustics, Speech and Signal Processing, 1975.

---

> > ### Author Rebuttal · Reviewer_Znsy · 2026-04-01
> >
> > Thanks for the answer.

---

> > > ### Author Response · Authors · 2026-04-07
> > >
> > > Thank you for your constructive review. We are glad the rebuttal helped address your concerns.

---

### Decision · Program_Chairs · 2026-04-30

**Decision:**

Accept (spotlight)

**Comment:**

This paper reformulates time series classification with noisy labels as a pairwise ranking problem using an Ideal Point Model (IPM). Instead of embedding data in a latent space, it learns a single reference time series and ranks samples based on their distance to it. To handle temporal distortions, the method employs elastic distances such as DTW and TWED, and introduces a differentiable soft-TWED for end-to-end optimization. The authors provide theoretical guarantees, including Lipschitz continuity and generalization bounds. Experiments on synthetic and real datasets demonstrate that the proposed approach outperforms L2-based IPM and ranking baselines.

The proposed method tackles the IPM framework for time series, which constitutes a solid contribution. The approach is simple yet supported by rigorous theoretical analysis. Since DTW has been widely studied across signal processing, data mining, and machine learning communities, this work is likely to have broad impact. Moreover, all concerns were adequately addressed during the rebuttal period, and the reviewers are satisfied with the responses. Therefore, I recommend this paper for acceptance to ICML 2026.